# FOLIAGE: A LATENT WORLD MODEL FOR ACCRETIVE SURFACE GROWTH

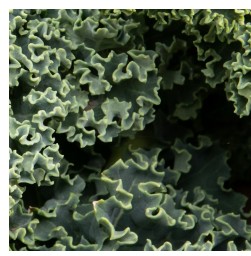
(a) Curly Kale

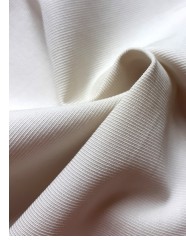
(b) Fabric

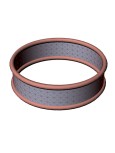
(c) Mesh at step 0

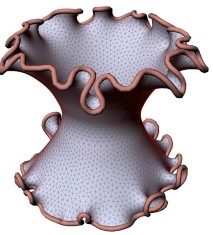
(d) Mesh at step 50

Figure 1: (a) Kale accretively grows bigger, gaining biomass and forming complex curls. (b) In contrast, passive sheets like cloth only deforms. In this work, FOLIAGE models the former, yielding stronger geometry understanding and cross-modal alignment. (c, d) SURF-GARDEN generates accretive growth sequences on which the SURF-BENCH suite provides evaluation.

## ABSTRACT

Accretive surfaces grow by adding material and changing rest metrics, producing emergent, complex, and changing morphologies. To study this phenomenon, we introduce FOLIAGE, a *geometry-centric* latent world model that infers a deployable state from heterogeneous, partial sensors and predicts its *action-conditioned* evolution. The perception stack aligns images, point clouds, and meshes through correspondence-constrained fusion and age features, then pools into global and young-region summaries that emphasize where change will occur next. Dynamics input act only on the latent, taking material coefficients and a horizon code to produce counterfactual roll-outs without entangling perception with control. Training-time physics guides representation via a target encoder that receives per-vertex energies and energy-gated message passing, while the deployable path relies solely on observable inputs. On our SURF-GARDEN data platform and the SURF-BENCH suite, FOLIAGE improves mesh topology classification by ∼3 pp, reduces dense-correspondence geodesic error by ∼10%, lifts cross-modal retrieval by ∼25% mAP@100, increases growth-stage recognition by ∼8 pp, lowers 5-step Chamfer by ∼20%, and cuts inverse-material error by ∼40% relative to strong baselines. Stress tests show graceful degradation under sensor loss, stable long-horizon roll-outs, and gains from train-only physics without test-time privileges. Code and datasets used in this study will be made publicly available upon publication to facilitate reproducibility and further research.

## 1 INTRODUCTION

Accretive surface growth adds material and alters rest metrics of thin shells (Fig. 1), inducing internal stress that differentially propagate to complex, global morphology evolution Coen et al. (2023); Efrati et al. (2009; 2017). This phenomenon underpins key frontier domains like smart material and 4D printing Gladman et al. (2016); Wang et al. (2024); van Manen et al. (2021). It is also present in living tissue such as plant and animal organs Liang & Mahadevan (2011); Huang et al. (2018). External actions through heat, light, and chemical signals can condition growth trajectories toward desirable shapes by changing the material property of the surface—without direct contact manipulation. Walden et al. (2023); Guo et al. (2021); Li et al. (2021). However, precise control

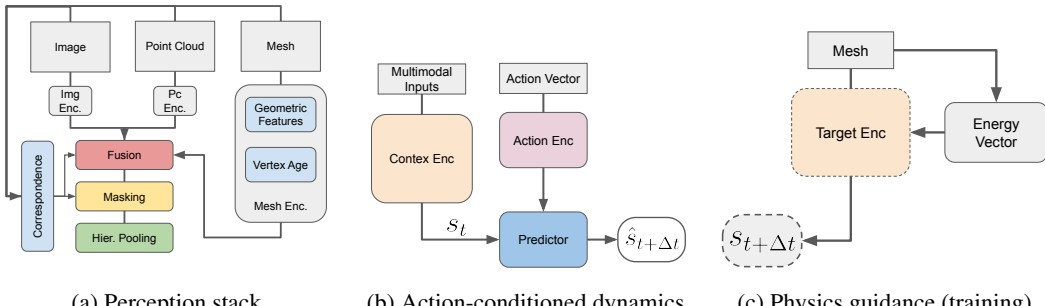

(a) Perception stack  (b) Action-conditioned dynamics  (c) Physics guidance (training)

Figure 2: In FOLIAGE: (a) a perception stack encodes sensors into latent $s_t$ (instantiated as context and EMA target encoders); (b) an action-conditioned predictor advances to $\hat{s}_{t+\Delta t}$; (c) (training only) a target encoder for physics-informed $s_{t+\Delta t}$.

and modeling is challenging because the internal physical forces that drive accretive growth is not externally observable; furthermore, growth sequence data is difficult to collect in both lab and natural settings Gallet et al. (2022); Ambrosi et al. (2019).

Existing methods largely fall into two families. Differentiable simulators expose solver internals and gradients, supporting parameter inference and control when models and discretizations are accurate and sensing is not the bottleneck Li et al. (2022); Hu et al. (2020; 2019). Video-oriented world models learn pixel dynamics end-to-end and are tuned for photometric prediction rather than surface geometry Hafner et al. (2019; 2023); Oh et al. (2015). The target regime here is different: geometry, not pixels, is the prediction object; sensing is multimodal and intermittent; and actions operate through material parameters. Moreover, practical sensing is heterogeneous and partial—images, point clouds, and occasionally partial meshes—creating a gap between raw observations and the geometry-centric state needed for prediction and control Tretschk et al. (2023); Mildenhall et al. (2020). Our objective is to infer latent a state from partial, multimodal observations and to predict how it evolves under actions on material coefficients Hu et al. (2021); Raissi et al. (2019); Ma et al. (2023).

**Design goals and problem formulation.** (G1) Accretion-aware perception that fuses heterogeneous sensors and emphasizes newly accreted regions. (G2) Action-conditioned latent dynamics that predict geometric evolution under material coefficients and time. (G3) Physics-guided representation learning that leverages train-only privileged signals while remaining deployable without test-time solvers. At time $t$, observations are $x_t \subseteq \{I_t, P_t, M_t\}$ for RGB images, point clouds, and a surface mesh. Actions $a_t \in \mathbb{R}^3$ parameterize material coefficients $[k_{\text{stretch}}, k_{\text{shear}}, k_{\text{bend}}]$. The model learns a latent state $s_t \in \mathbb{R}^d$ and a predictor $P_\theta$ such that for horizons $\Delta t$, $s_{t+\Delta t} \approx P_\theta(s_t, a_t, \Delta t)$.

**Model overview.** FOLIAGE (Fig. 2) is a latent world model Ha & Schmidhuber (2018); Hafner et al. (2019) targeted at accretive surfaces under partial sensing and material control. A perception stack maps available sensors into a shared token set with correspondence-based fusion and an accretion-aware mesh pathway with vertex birth-time tags. Hierarchical pooling forms a compact state comprising a global summary and a young-region summary. Actions enter only in the dynamics through an action token; a lightweight Transformer predicts $\hat{s}_{t+\Delta t}$ from $(s_t, a_t, \Delta t)$, preserving counterfactual semantics. During training, a target encoder receives per-vertex energies and applies energy-gated message passing to shape supervision; deployment uses observable inputs only.

**Data and benchmark.** The SURF-GARDEN platform produces sequences in which surfaces grow and deform under internal stress and material response while recording exact correspondences across sensors, enabling multimodal supervision and counterfactual branching. An accompanying evaluation suite, SURF-BENCH, defines tasks spanning geometry understanding, cross-modal alignment, rollouts, and inverse materials.

**Contributions.** (1) A geometry-centric latent world model that fuses images, point clouds, and meshes and predicts action-conditioned evolution under partial sensing. (2) An accretion-aware perception stack with vertex birth-time tags, correspondence-driven fusion, and hierarchical pooling emphasizing young regions. (3) A physics-guided training branch that uses per-vertex energies via energy-gated message passing, with deployment relying only on observable inputs. (4) A data platform and an evaluation suite for accretive surface dynamics.

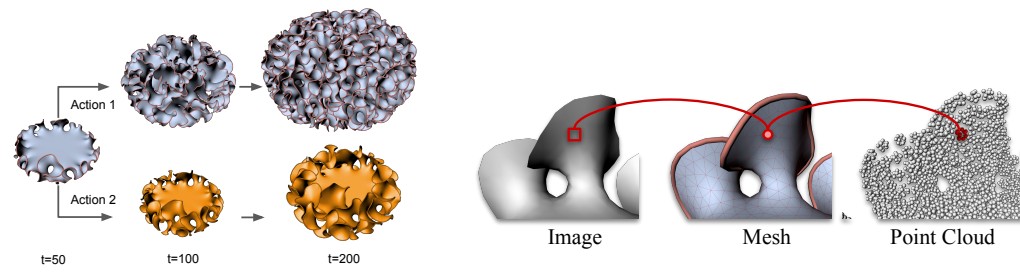

(a) Counterfactual branching          (b) Correspondence registration

Figure 3: SURF-BENCH explores (a) a rich action space with (b) fine-grained correspondences.

## 2    RELATED WORK

**Latent world models for control.** Latent world models learn compact states and their dynamics from high-dimensional observations to support prediction, planning, and control Hafner et al. (2019); Ha & Schmidhuber (2018); Lee et al. (2020). Most systems are designed for image-centric tasks, where the latent is optimized for reward prediction or pixel reconstruction and actions represent task controls in relatively rigid scenes Hafner et al. (2023); Oh et al. (2015). These approaches typically rely on decoders or value models and seldom represent geometry explicitly; decoder-free variants remain predominantly pixel-based with limited support for heterogeneous sensors Mildenhall et al. (2020); Kerbl et al. (2023); Jaegle et al. (2021). In contrast, this work targets **geometry as state** and fuses images, point clouds, and meshes through correspondence, while **G2** introduces action-conditioned dynamics tied to material coefficients Xu et al. (2022); Teed & Deng (2021). The representation is built to support downstream geometric competence under partial sensing, without obligating image synthesis or reconstruction.

**Learning physical dynamics.** Differentiable simulators expose solver internals and gradients for inverse problems and control when discretizations and constitutive laws are accurate, but they inherit solver stability and modeling errors at inference Hu et al. (2021); Li et al. (2022); Hu et al. (2019; 2020). Neural surrogates and graph-based simulators advance states directly via learned message passing or operator learning, typically on fixed or slowly varying meshes/particles with explicit state supervision Sanchez-Gonzalez et al. (2020); Pfaff et al. (2021). Both strands largely optimize *explicit* physical states and assume stable connectivity, which complicates growth regimes where new surface elements are spawned Pfaff et al. (2021); Sanchez-Gonzalez et al. (2020); Li et al. (2022). Our approach places dynamics guidance at **training time**: FOLIAGE builds a **growth-aware latent** via correspondence-driven fusion (G1), then learns **action-conditioned** evolution tied to material coefficients (G2), while remaining **solver-free at deployment** (G3). We observe that decoupling representation learning from solver fidelity improves robustness under multimodal, intermittent sensing and topology change.

**Multimodal geometric learning and correspondence.** Cross-modal representation learning aligns images with 3D geometry to support retrieval and correspondence Li et al. (2023); Zhu et al. (2022); Liu & et al. (2023), while mesh correspondence methods focus on accurate matching across surfaces Ovsjanikov et al. (2012); Melzi et al. (2019); Donati et al. (2020). These lines usually assume static or pre-meshed geometry and are not optimized jointly with action-conditioned temporal dynamics or robustness to missing modalities. FOLIAGE strengthen **correspondence-driven fusion** Newcombe et al. (2015); Innmann et al. (2016); Bozic et al. (2020) with a dynamics predictor so that the latent is effective for both static geometric tasks and forecasting under material-driven evolution (**G1+G2**). Structured masking and hierarchical pooling increase robustness to incomplete sensing and emphasize recently accreted regions Chen et al. (2023a); Yu et al. (2022); Liu et al. (2022).

## 3    SURF-BENCH: A DATA PLATFORM FOR ACCRETIVE SURFACE GROWTH

To train our model and motivate further investigation in accretive growth, we build the SURF-GARDEN pipeline (Fig. 3) which generates physically simulated mesh sequences with precisely aligned multi-view RGB and LiDAR-style observations. The resulting dataset contains $7,200$

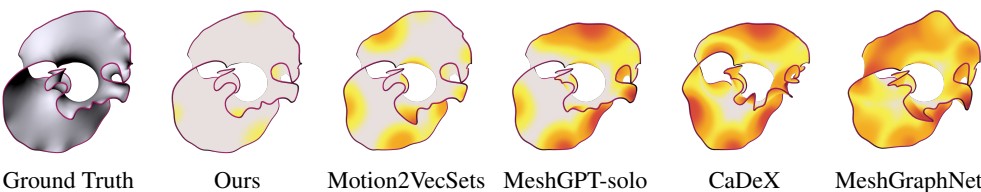

Ground Truth    Ours    Motion2VecSets    MeshGPT-solo    CaDeX    MeshGraphNets

Figure 4: Mesh predictions over 10 time steps under control actions. Errors are illustrated by a color gradient. FOLIAGE's predictions closely resemble the ground truth while baselines' rapidly deviate.

branched sequences of $400$ frames each, with vertex counts growing from $\sim 20$ to $\sim 10^5$, and an $8\!:\!1\!:\!1$ train/val/test split.

**Simulator.** We simulate a thin elastic shell with membrane and bending energies Tamstorf & Grinspun (2013), introduce growth via metric accretion that expands rest lengths and induces non-Euclidean curvature. We remesh to maintain quality and prevent self-intersections across diverse genera.

**Counterfactual branching.** After a shared 50-frame prefix, each sequence branches under modified controls. Mesh element identifiers persist through connectivity updates, keeping geometry and indices aligned across branches and enabling supervised counterfactual roll-outs from a shared past without further simulator calls. To prevent split leakage, branches that descend from the same pre-branch 'parent' trajectory are assigned to the same split; no parent or its branches are split across train/val/test.

**Multimodal correspondence.** Per frame, we render multi-view RGB and a LiDAR-style point cloud. Pixels and points carry their originating mesh elements, and camera parameters are fixed per trajectory, yielding exact cross-modal and temporal correspondences for the encoders.

## 4  FOLIAGE: A LATENT WORLD MODEL FOR ACCRETIVE SURFACES

**System overview and goals.** FOLIAGE targets accretive surface dynamics under heterogeneous and intermittent sensing by learning a geometry-centric latent state and its action-conditioned evolution (Alg. 4.1). It has three goals: **G1** accretion-aware perception that fuses images, point clouds, and meshes and emphasizes newly accreted regions; **G2** action-conditioned latent dynamics that predict geometry under material controls while keeping perception observational; **G3** physics-guided representation learning that uses train-only privileged signals without requiring a solver at deployment.

### 4.1  Accretion-aware perception (G1).

The perception stack (Fig. 2a) yields a latent $s_t \in \mathbb{R}^d$ that (i) respects growth-driven connectivity changes, (ii) tolerates heterogeneous and intermittent sensing, and (iii) highlights recently accreted regions predictive of near-term evolution. It comprises three stages: modality encoders, correspondence-driven fusion, and hierarchical pooling. All components in this section use observable inputs only; privileged physics appears later in the target branch.

**Modality encoders.** Let $x_t \subseteq \{I_t, P_t, M_t\}$ be the active sensors. Each sensor is mapped to tokens in a shared space: $\mathcal{T}_I = \{\mathbf{p}_k\}$, $\mathcal{T}_P = \{\mathbf{q}_k\}$, and $\mathcal{T}_M = \{\mathbf{r}_v\}$; missing sensors contribute empty sets and downstream modules operate on the union $\mathcal{U}_t = \mathcal{T}_I \cup \mathcal{T}_P \cup \mathcal{T}_M$ without branching. The mesh pathway Sharp et al. (2022); Thorpe et al. (2022) encodes growth-driven connectivity in which vertices are spawned as area increases (Fig. 5). Each vertex carries geometric features and a birth-time tag $\tau(v)$; an age feature marks recently spawned regions and is injected before message passing. Image and point tokens lack intrinsic age and receive an age proxy via correspondences (Fig. 3).

**Correspondence-driven fusion.** Tokens interact over a sparse heterogeneous graph with edges from pixels to intersected mesh elements, points to nearest mesh vertices, and mesh-mesh adjacency. Fusion uses type-aware attention constrained to these edges with simple geometric biases (e.g., barycentric confidence or distance) to privilege reliable correspondences, allowing texture to refine mesh tokens and points to supply metric anchors without dense all-to-all interactions.

**Robustness under partial sensing.** A structured masking scheme enforces robustness: random token masking per active modality, paired masking of neighbors along correspondence edges near masked tokens, and occasional modality-level dropout (applied stochastically during training).

---

**Algorithm 1** Training and inference for FOLIAGE

1: **Input:** dataset $\mathcal{D}$ of sequences of observations $x_t$, actions $a_t$, privileged physics $w_t$
2: **Parameters:** context encoder $E_{\text{ctx}}$, predictor $P$, EMA decay $\alpha$, $E_{\text{tar}}$
3: **Training**
4:    Sample $(x_t, x_{t+\Delta t}, a_t, w_{t+\Delta t})$ from $\mathcal{D}$, with $\Delta t \sim \text{Uniform}\{1, \ldots, 20\}$
5:    $s_t \leftarrow E_{\text{ctx}}(x_t)$            ▷ accretion-aware perception from observables (Sec. 4.1)
6:    $\hat{s}_{t+\Delta t} \leftarrow P(s_t, a_t, \Delta t)$            ▷ action-conditioned latent dynamics (Sec. 4.2)
7:    $s_{t+\Delta t} \leftarrow E_{\text{tar}}(x_{t+\Delta t}, w_{t+\Delta t})$            ▷ teacher sees future state w/ physics (Sec. 4.3)
8:    $L_{\text{pred}} \leftarrow \|\hat{s}_{t+\Delta t} - s_{t+\Delta t}\|_2^2$
9:    Update $E_{\text{ctx}}$ and $P$ using $\nabla L_{\text{pred}}$            ▷ no gradients through $E_{\text{tar}}$ or $w_{t+\Delta t}$
10:    $E_{\text{tar}} \leftarrow \alpha E_{\text{tar}} + (1 - \alpha)E_{\text{ctx}}$            ▷ EMA teacher update
11: **Inference (given $x_t, a_t, \Delta t$):**
12:    $s_t \leftarrow E_{\text{ctx}}(x_t)$; $\hat{s}_{t+\Delta t} \leftarrow P(s_t, a_t, \Delta t)$
13:    **return** $(s_t, \hat{s}_{t+\Delta t})$

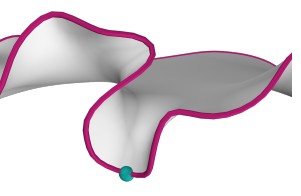 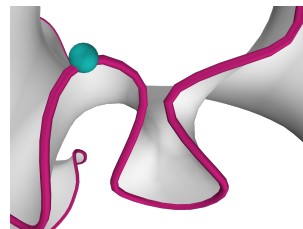 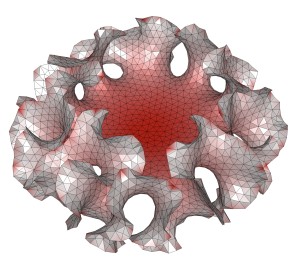

(a) Vertex on mesh at t=0            (b) Same vertex by t=7            (c) Mesh colored by vertex age

Figure 5: Accretive growth complicates learning consistent features over time. (a) a vertex (green) that began at a concave region quickly ends up at a convex area (b) as the mesh evolves. (c) tracking the age of vertices added to the mesh illuminates this dynamic: in actively growing areas, young vertices (white) quickly emerge between old vertices (red).

**Age propagation to non-mesh tokens.** Non-mesh tokens inherit an age proxy from neighboring mesh tokens through correspondence edges using a conservative rule that assigns the minimum neighboring age; the proxy is used for pooling only.

**Hierarchical pooling and state formation.** Pooling compresses $\mathcal{U}_t$ into a fixed-size state using two summaries: a global token summary and a young-region summary computed over tokens with small age. The two summaries are concatenated and linearly projected to form $s_t$.

**Interfaces and invariances.** The perception stack is permutation-invariant within token sets, tolerant to missing modalities by construction, and stable to growth-driven mesh refinement. $s_t$ is the sole geometric state for downstream action-conditioned prediction; separating perception from actions preserves counterfactual semantics and the masking scheme prevents trivial copying.

## 4.2 Action-conditioned latent dynamics (G2).

The predictor (Fig. 2b) advances the latent state under material control. Given $s_t$, $a_t$, and horizon $\Delta t$, it produces $\hat{s}_{t+\Delta t}$. Actions enter only here, keeping perception observational and enabling counterfactual roll-outs from a fixed $s_t$.

**Inputs, conditioning, and training.** Inputs are the state $s_t \in \mathbb{R}^d$, material coefficients $a_t \in \mathbb{R}^3$ encoded by a small MLP into an action token, and a horizon code $\phi(\Delta t)$. The concatenated vector $[s_t \| \mathbf{a}_t \| \phi(\Delta t)]$ is projected and updated by a compact Transformer to yield $\hat{s}_{t+\Delta t}$. Multi-horizon training samples $\Delta t$ uniformly from $\{1, \ldots, 20\}$ and minimizes $\mathcal{L}_{\text{pred}} = \|\hat{s}_{t+\Delta t} - s_{t+\Delta t}\|_2^2$ where targets come from a target encoder (exponential-moving-average copy of the context encoder), stabilizing supervision across horizons.

**Null action and stability.** When controls are unavailable, a learned null-action embedding replaces $\mathbf{a}_t$. Multi-horizon sampling and a bottlenecked action path mitigate long-horizon drift and over-conditioning while preserving sensitivity to geometry encoded in $s_t$.

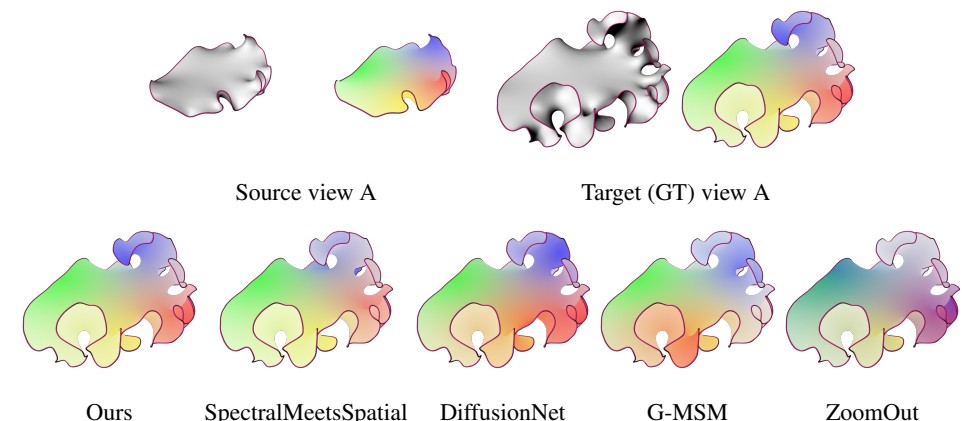

Figure 6: FOLIAGE accurately tracks features in the dense correspondence tasks as baselines degrade over significant changes in surface morphology which occur during accretive growth.

### 4.3 PHYSICS-GUIDED REPRESENTATION (G3).

Growth is driven by internal stress and material response, which are not observable at deployment. During training only, the simulator provides per-vertex membrane and flexural energies that summarize these drivers. G3 uses these privileged signals to bias the representation toward regions likely to evolve under control, while the deployed system relies solely on observable inputs.

**Architecture.** Two instantiations of the perception stack (Fig. 2a) are maintained: a context encoder $E_{ctx}$ that consumes only observables (Fig. 2b), and a target encoder $E_{tar}$ that additionally receives per-vertex energies during training (Fig. 2c). $E_{tar}$ supervises the predictor and is updated as an exponential moving average Tarvainen & Valpola (2017) of $E_{ctx}$, preserving symmetry and avoiding a test-time dependency on privileged values.

**Energy-gated message passing and auxiliary alignment.** In $E_{tar}$, energies modulate the early mesh message passing so that propagation is amplified near high-energy vertices and remains standard elsewhere. Gating is bounded and confined to the first propagation step for stability. $E_{ctx}$ has no gating. Target mesh tokens also predict normalized energies with a lightweight regression loss, encouraging features that correlate with stress without changing the deployed path.

**Training objective, leakage control, and safeguards.** The predictor is trained to match the target latent $s_{t+\Delta t} = E_{tar}(x_{t+\Delta t})$ using $\mathcal{L}_{pred}$, with a simple variance–covariance regularizer to avoid collapsed representations; weights are shared across branches except for the gated inputs in $E_{tar}$. Privileged energies are provided only to $E_{tar}$. Energies are treated as constants (no gradients). $E_{tar}$ is updated via EMA rather than by backpropagating prediction gradients through privileged inputs. Bounded gating and a single gated step prevent over-amplification; EMA coupling keeps $E_{tar}$ close to $E_{ctx}$, reducing sensitivity to simulator idiosyncrasies and preserving deployment behavior.

## 5 SURF-BENCH EXPERIMENTS

We assemble the SURF-BENCH suite composed of six tasks (T1–T6) and four stress tests (S1–S4), aligned with the model goals: R1 (G1) geometry and multimodal grounding (T1, T5, T6); R2 (G2) action-conditioned dynamics (T2, T4); R3 (G3) physics-guided representation (T3); R4 robustness and generalization (S1–S4). Success criteria: the latent supports classification, regression, correspondence, retrieval, and roll-outs under partial sensing, and remains counterfactually coherent so that identical $s_t$ can condition distinct futures under different $a_t$. For each downstream task, a specialized critic head ingesting the learned latent is used. FOLIAGE does not receive modalities not present in the task input (e.g. image for dense correspondence).

**1. Geometry, correspondences, and cross-modal alignment under partial inputs.** Tasks T1 (global shape), T5 (image–point retrieval), and T6 (dense correspondence) assess whether the latent is geometry-faithful and sensor-robust. Baselines for T1 include MeshCNN Hanocka et al. (2019),

| **T1: Topology Classification** | | **T2: Inverse Material Reg.** | | **T3: Growth-Stage Recog.** | |
|---|---|---|---|---|---|
| | Accuracy↑ | | MAE↓ | | Balanced Acc.↑ |
| MeshCNN | 0.88 | NeuralClothSim | 0.060 | SVFormer | 0.67 |
| DiffusionNet | 0.92 | DiffPD | 0.058 | VideoMAE-v2 | 0.68 |
| Adaptive-PH | 0.94 | BDP | 0.055 | TimeSformer | 0.69 |
| ETNN | 0.94 | DiffCloth | 0.053 | VideoMamba | 0.71 |
| **Ours** | **0.97** | **Ours** | **0.038** | **Ours** | **0.79** |
| **T4: Mesh Forecasting** | | **T5: Cross-Modal Retrieval** | | **T6: Dense Correspondence** | |
| | Chamfer↓/Vertex Drift↓ | | mAP@100↑ | | Geodesic Err.↓ |
| MeshGraphNets | 0.065/4133 | CrossPoint | 0.42 | ZoomOut | 4.2 |
| CaDeX | 0.052/2261 | CLIP2Point | 0.43 | DiffusionNet | 3.8 |
| MeshGPT-solo | 0.045/2540 | ULIP-2 | 0.46 | G-MSM | 3.6 |
| Motion2VecSets | 0.038/1721 | PointCLIPv2 | 0.48 | SpectralMeetsSpatial | 3.2 |
| **Ours** | **0.030/1044** | **Ours** | **0.60** | **Ours** | **2.8** |

Table 1: FOLIAGE's learning outcomes apply to a diverse range of downstream tasks (T1 T6).

DiffusionNet Sharp et al. (2022), Adaptive-PH Nishikawa et al. (2023), and ETNN Battiloro et al. (2023); for T5 include CrossPoint Zhang et al. (2021), CLIP2Point Zhang et al. (2023), ULIP-2 Li et al. (2023), and PointCLIPv2 Zhu et al. (2022); and for T6 include ZoomOut Melzi et al. (2019), DiffusionNet Sharp et al. (2022), G-MSM Eisenberger et al. (2023), and SpectralMeetsSpatial Cao et al. (2024a). Protocols use identical sensor subsets and correspondence supervision when applicable.

**2. Action-conditioned dynamics and stability beyond training horizons.** Tasks T2 (inverse material estimation) and T4 (counterfactual roll-outs) evaluate control sensitivity versus passive prediction. Baselines for T2 include differentiable-physics identification with DiffPD Hu et al. (2021) and BDP Gong et al. (2024), and neural simulators such as NeuralClothSim Kairanda et al. (2024) and DiffCloth Li et al. (2022). Baselines for T4 include forecasters MeshGraphNets Pfaff et al. (2021), CaDeX Lei & Daniilidis (2022), MeshGPT-solo Siddiqui et al. (2024), and Motion2VecSets Cao et al. (2024b); these operate as passive forecasters unless otherwise specified.

**3. Physics-guided training and anticipation of change.** Task T3 (growth-stage recognition) tests whether training-time privileged signals improve sensitivity to regions about to evolve, without any privileged inputs at deployment. Baselines span per-modality classifiers (SVFormer Chen et al. (2023b), TimeSformer Bertasius et al. (2021)), fusion or masked encoders (VideoMAE-v2 Wang et al. (2023)), and temporal backbones (VideoMamba Li et al. (2024)).

**4. Robustness and generalization challenges.** Stress tests include sensor-subset robustness (S1), zero-shot alignment (S2), long-horizon stability (S3), and physics ablation (S4). Baselines for S1 include VideoMAE-v2 Wang et al. (2023) and PiMAE Chen et al. (2023a); for S2 include CrossPoint Zhang et al. (2021), CLIP2Point Zhang et al. (2023), ULIP-2 Li et al. (2023), and PointCLIPv2 Zhu et al. (2022); for S3 include FMNet Rodolà et al. (2017), MeshGraphNets Pfaff et al. (2021), MeshGPT-solo Siddiqui et al. (2024), CaDeX Lei & Daniilidis (2022), INSD Sang et al. (2025), and Motion2VecSets Cao et al. (2024b); and for S4 include classical pooling and graph backbones $\mu$Pool Zaheer et al. (2017) and GCN Kipf & Welling (2017). Ablations remove individual inductive components (GCF, XPM, APE, energy signals, EGMP) to isolate their contributions.

# 6 RESULTS

## 6.1 CORE TASKS

Tab. 1 reports performance on the SURF-BENCH core tasks.

**Geometry Understanding (T1, T6).** Across the two hardest purely geometric tests—classifying mesh genus and recovering dense correspondences—FOLIAGE adds a consistent ∼3 pp of accuracy and cuts geodesic error by ≈ 10% versus the strongest recent baselines. The gain follows from treating geometry as part of a world state: age features disambiguate birth and death of vertices, and the global/young-region summaries preserve scene-level context when solving functional maps. Spectral or diffusion descriptors that view each shape in isolation lack this temporal context.

**S1 Sensor-Subset Robustness** (Balanced Acc.↑)

| | Rich | Typical | Sparse | Noisy |
|---|---|---|---|---|
| VideoMAE-v2 | 0.74(4) | 0.68(6) | - | 0.62(7) |
| PiMAE | 0.76(5) | 0.70(6) | 0.67(6) | 0.63(6) |
| ULIP-2 | 0.78(4) | 0.72(5) | 0.71(5) | 0.66(6) |
| CLIP2Point | 0.74(5) | 0.65(5) | 0.68(6) | 0.60(6) |
| **Ours** | **0.80(3)** | **0.78(4)** | **0.74(4)** | **0.74(4)** |

**S2 Zero-Shot Img.-Pc. Retrieval** (mAP@100↑)

| | $I{\to}P$ | $P{\to}I$ |
|---|---|---|
| CrossPoint | 0.18 | 0.16 |
| CLIP2Point | 0.20 | 0.19 |
| PointCLIPv2 | 0.23 | 0.21 |
| ULIP-2 | 0.22 | 0.23 |
| **Ours** | **0.38** | **0.36** |

**S3 Long-Horizon Latent Roll-outs** (Chamfer↓)

| | 5 | 10 | 20 | 40 |
|---|---|---|---|---|
| FMNet | 0.042 | 0.053 | 0.092 | 0.136 |
| MeshGraphNets | 0.036 | 0.057 | 0.088 | 0.120 |
| MeshGPT-solo | 0.027 | 0.052 | 0.089 | 0.110 |
| CaDeX | 0.028 | 0.040 | 0.068 | 0.990 |
| INSD | 0.029 | 0.035 | 0.055 | 0.890 |
| Motion2VecSets | 0.022 | 0.029 | 0.045 | 0.075 |
| **Ours** | **0.016** | **0.025** | **0.028** | **0.047** |

**Architectural Ablation** (mean (stdev))

| | Topo↑ | MAE↓ | Chamfer↓ | mAP↑ |
|---|---|---|---|---|
| w/o GCF | 0.960(2) | 0.040(2) | 0.033(2) | 0.46(10) |
| w/o XPM | 0.975(2) | 0.038(2) | 0.031(2) | 0.55(10) |
| w/o APE | 0.951(3) | 0.042(3) | 0.036(3) | 0.59(10) |
| $\mu$Pool | 0.960(2) | 0.039(2) | 0.034(2) | 0.57(10) |
| GCN | 0.932(4) | 0.045(3) | 0.038(3) | 0.53(20) |
| w/o EGMP | 0.964(2) | 0.048(2) | 0.030(2) | 0.60(10) |
| **Ours** | **0.958(2)** | **0.035(2)** | **0.028(2)** | **0.63(10)** |

**S4 Energy-Signal Ablation**

| | MAE↓ |
|---|---|
| w/o Energy-All | 0.060(3) |
| w/o Energy-Aux | 0.048(2) |
| **Our Full** | **0.035(2)** |

**Capacity / Compute (Training)**

| | Params | GPU-h |
|---|---|---|
| GCN | 27 | 11 |
| w/o XPM | 39 | 18 |
| **Our Full** | **41** | **19** |

Table 2: FOLIAGE degrades gracefully in stress tests (S1-S4). Ablations illuminate the impact of our design choices on performance and cost.

| Method | GrowliFlower (T3) Balanced Acc. ↑ | Pheno4D (T6) Geodesic. Err. ↓ | Crops3D (T6) Geodesic. Err. ↓ |
|---|---|---|---|
| FOLIAGE (Frozen) | 0.72 | 3.55 | **4.45** |
| FOLIAGE (Few-shot) | 0.75 | – | – |
| FOLIAGE (Light-ft) | **0.77** | **3.25** | – |
| VideoMAE-v2 | 0.70 | – | – |
| TimeSformer | 0.68 | – | – |
| VideoMamba | 0.69 | – | – |
| SVFormer | 0.66 | – | – |
| DiffusionNet | – | 3.90 | 4.80 |
| G-MSM | – | 4.60 | 5.50 |
| SpectralMeetsSpatial | – | 5.05 | 5.95 |
| ZoomOut | – | 5.70 | 6.70 |

| Method | D-FAUST (T6) Geodesic. Err. ↓ | CAPE (T6) Geodesic. Err. ↓ |
|---|---|---|
| FOLIAGE (Frozen) | 3.6 | 4.4 |
| FOLIAGE (Light-ft) | **3.1** | **3.8** |
| DiffusionNet | 3.8 | 4.7 |
| G-MSM | 4.5 | 5.4 |
| SpectralMeetsSpatial | 4.9 | 5.8 |
| ZoomOut | 5.6 | 6.6 |

Table 3: Real-world and cross-domain transfer. *Top*: sim-to-real evaluation on real plant datasets. We test growth-stage recognition (T3) on GrowliFlower Kierdorf et al. (2023) and dense correspondence (T6) on Pheno4D Schunck et al. (2021) and Crops3D Zhu et al. (2024). *Bottom*: cross-domain generalization to dynamic human bodies (D-FAUST Bogo et al. (2017), CAPE Ma et al. (2020)) using the T6 correspondence head. "Frozen" keeps the encoder fixed; "Few-shot" trains only a linear probe with 10 labeled sequences per class; "Light-ft" additionally finetunes the last encoder block.

**Physical Parameter Inference (T2).** Regressing bending modulus from a single RGB view, differentiable-physics identification beats vision-only CNNs, yet FOLIAGE reduces error by $\approx 40\%$.

| Method | Clean | Corr-noise | Occlusion | Drift | Drift+SLAM |
|---|---|---|---|---|---|
| | | mAP@100 ↑ (T5) | | | |
| FOLIAGE (full) | **0.60** | **0.56** | **0.51** | **0.52** | **0.59** |
| FOLIAGE (w/o GCF) | 0.46 | 0.45 | 0.43 | 0.44 | 0.46 |
| ULIP-2 | 0.46 | 0.49 | 0.40 | 0.41 | 0.42 |
| PointCLIPv2 | 0.48 | 0.47 | 0.39 | 0.40 | 0.41 |
| CLIP2Point | 0.43 | 0.45 | 0.37 | 0.38 | 0.39 |
| CrossPoint | 0.42 | 0.43 | 0.35 | 0.36 | 0.37 |

Table 4: Robustness under partial sensing and imperfect correspondences. We stress-test cross-modal retrieval (T5) with correspondence corruption: Corr-noise randomly rewires 20% of pixel/point→mesh edges within local geodesic neighborhoods and drops 25% of remaining cross-modal edges; Occl. masks 50% of RGB tokens with contiguous rectangles and co-masks linked 3D tokens; Drift injects moderate camera pose/intrinsics jitter, and Drift+SLAM re-estimates poses with standard visual-SLAM. "w/o GCF" removes correspondence-constrained fusion.

While the model does not simulate mesh states, physics guidance in training shapes a latent on which a small head suffices for inverse material estimation.

**Growth Perception & Prediction (T3, T4).** Video transformers detect growth mainly through pixel motion; FOLIAGE leverages the young-region summary and improves stage recognition by ∼8 pp. Rolling the same latent forward reduces 5-step Chamfer error by roughly one fifth while avoiding spurious vertex explosions. The predictor anticipates localized edits that realize accretion, not only smooth deformations.

**Cross-Modal Grounding (T5).** Correspondence-constrained fusion ties pixel tokens to their source vertices, narrowing cross-modal gap and yielding a 25% relative boost in mAP@100 over the strongest retrieval baseline. In the multimodal world state, geometry and appearance cohabit the same coordinate frame—which improves cross-modal search without task-specific retraining.

## 6.2 STRESS TESTS AND EXTENDED STUDIES

We freeze the encoder and probe four settings that stress modality resilience, cross-modal alignment, long-horizon stability, and physics supervision (Tab. 2). We also test real-world and cross-domain generalization (Tab. 3) and reliance on correspondences (Tab. 4).

**Modality-Robust Inference (S1, S2).** Across Rich (RGB+LiDAR+mesh), Typical (RGB-only), Sparse (LiDAR-only), and Noisy (masked RGB) settings, FOLIAGE leads in balanced accuracy and degrades gracefully. The strongest baseline (ULIP-2 Li et al. (2023)) drops seven points from Rich to Sparse; FOLIAGE drops six and holds ground under noisy RGB, reflecting sensor elasticity from structured masking and correspondence fusion. The same design drives zero-shot retrieval: FOLIAGE reaches $0.38/0.36$ mAP@100 (image–point), nearly doubling CrossPoint and leading CLIP2Point Zhang et al. (2023), PointCLIPv2 Zhu et al. (2022), and ULIP-2 by 14–16 pp.

**Predictive Fidelity & Physics Signals (S3, S4).** Extrapolating further in time, FOLIAGE maintains temporal coherence as physics-based baselines degrade. Removing privileged stretch/bend energies increases inverse-material MAE on $k_{\text{bend}}$ from 0.035 to 0.060; keeping EGMP while dropping the auxiliary head recovers part of this gap, indicating that detached physics cues at training time improve deployable accuracy without test-time privileges.

**Generalization.** On natural plant growth (GrowliFlower Kierdorf et al. (2023)), a frozen linear probe attains 0.72 balanced accuracy and 0.77 with light finetuning, outperforming video baselines. For dense correspondence on Pheno4D Schunck et al. (2021) and Crops3D Zhu et al. (2024), we drop oracle links and use anchored ICP Besl & McKay (1992) with pruning and pose smoothing; frozen FOLIAGE yields 3.55/4.45 geodesic error and remains best among correspondence methods, improving to 3.25 on Pheno4D with light finetuning. Finally, on dynamic-human benchmarks (D-FAUST Bogo et al. (2017), CAPE Ma et al. (2020)), the same correspondence head is competitive when frozen and leads after light finetuning, indicating that the latent captures broadly useful dynamic-3D structure rather than simulator-specific artifacts.

**Imperfect Correspondence.** Under pixel/point-mesh occlusion, and camera drift, FOLIAGE retains a clear lead over baselines. Removing correspondence-constrained fusion collapses toward correspondence-free baselines, suggesting that correspondences are helpful but are not brittle. With moving cameras, SLAM-based pose correction nearly restores clean performance (0.59), bounding dependence on oracle associations in realistic capture regimes.

## 6.3 ABLATION STUDIES

In Tab. 2, each variant disables a single component, retrains for the same 20 GPU-h, and is scored on topology accuracy, material MAE, 5-step Chamfer, and retrieval mAP (mean $\pm$ stdev, 3 seeds). Three design choices emerge as most impactful:

**(i) Geometry-aware fusion.** Removing correspondence-constrained fusion minimally affects topology (–1 pp) but reduces retrieval by 14 pp. Replacing structured masking with random masking degrades every metric, confirming the need to simulate sensor dropout so multimodality provides flexibility rather than failure under missing inputs.

**(ii) Temporal encoding.** Without age features, material MAE and Chamfer increase (e.g., $+0.004$ cm) while retrieval remains flat, suggesting that age primarily encodes growth dynamics, not appearance. Replacing hierarchical pooling with mean pooling Zaheer et al. (2017) hurts all tasks, indicating that separating global and young-region summaries to capture multiscale dynamics is important.

**(iii) Capacity and physics-informed gating.** A size-matched 10-layer GCN Kipf & Welling (2017) (27M params, 11 GPU-h) trails the full model (41M, 19 GPU-h) by 4–7 pp, suggesting gains are architectural rather than purely parametric. Removing EGMP while keeping the auxiliary loss nearly doubles material MAE, underscoring the general benefit of train-time physics guidance.

## 6.4 ANALYSIS

**Comparison with simulator-centric pipelines.** Explicit simulators excel when full states and accurate discretizations are available, but deployment often lacks solver access and faces partial sensing. Across T2 and T4, the geometry-centric latent paired with action-conditioned dynamics yields lower inverse-material error and more stable roll-outs than pipelines that rely on differentiable gradients at inference. S4 shows that privileged energies improve supervision yet are unnecessary at test time: training-only physics reduces $k_{\text{bend}}$ MAE substantially while preserving the deployable interface. The broader guidance is to use physics to shape representation during training, keep actions out of perception to preserve counterfactual semantics, and evaluate success on geometry- and control-centric metrics rather than pixel error.

**Comparison with video-centric encoders.** Video backbones optimized for photometric objectives transfer poorly to cross-modal geometry tasks and under sensor loss. On T5 and S2, correspondence-constrained fusion and a geometry-centric state deliver large mAP gains and zero-shot alignment that video encoders do not match; on T1/T6, temporally informed geometry (age features + young-region pooling) lowers classification and correspondence errors without relying on dense appearance cues. Under S1, structured masking trains for sensor elasticity, so the model degrades smoothly from Rich to Sparse regimes where video methods drop sharply. The broader guidance is to treat geometry as the state, fuse modalities through explicit correspondences, and encode growth locality directly in the state; these choices yield generalizable improvements in retrieval, correspondence.

## 7 CONCLUSION

FOLIAGE treats *geometry as state* and couples correspondence-driven perception with *action-conditioned* latent dynamics, using physics only at training time to shape targets. This design yields counterfactual roll-outs, robust cross-modal grounding, and improved geometric competence under partial sensing, while avoiding dependence on solver access or pixel reconstruction. Results on SURF-BENCH indicate consistent gains in topology, correspondence, retrieval, growth recognition, roll-out stability, and inverse materials. The combination of geometry-centric state, explicit cross-modal correspondences, and train-only physics guidance provides a compact and deployable recipe for modeling growing surfaces and, more broadly, for physical world models operating under heterogeneous sensing and control.

## 8 ETHICS, LLM, AND REPRODUCIBILITY STATEMENT

We have read, acknowledged, and adhered to the ICLR Code of Ethics. Large Language Models (ChatGPT) were used exclusively to improve the clarity and fluency of English writing following the completion of the draft by authors. They were not involved in research ideation, experimental design, data analysis, or interpretation. The authors take full responsibility for all content. We provide further details on architectural specifications, hyperparameters, ablations, and metric definitions are documented in the appendix. We believe that these materials enable independent reproduction of the reported results, and we will release the source code and pretrained models upon acceptance to further facilitate reproducibility and research.

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

# A APPENDIX

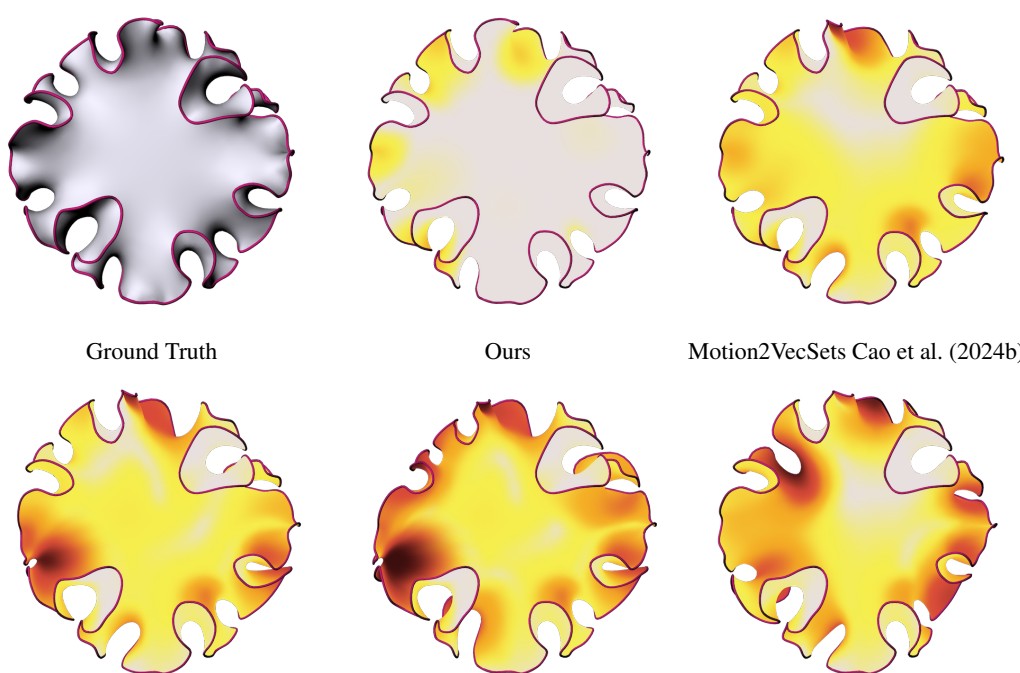



Ground Truth      Ours      Motion2VecSets Cao et al. (2024b)

MeshGPT-solo Siddiqui et al. (2024)    CaDeX Lei & Daniilidis (2022)    MeshGraphNets Pfaff et al. (2021)



Figure 7: Mesh predictions on SURF-BENCH. $\Delta t = 18$, action = $[0.1, 0.3, 0.01]$.

## A.1 SURF-GARDEN DETAILS

In SURF-GARDEN, surfaces undergo *metric accretion*, where new material is added, and deform like thin elastic shells. Energy-based simulations produce FEM-quality meshes with two sensor projections per frame. With rich physics, precise cross-modal alignment, and topological diversity, SURF-GARDEN offers a well-rounded addition to accretive surface growth research.

### A.1.1 COUNTERFACTUAL PHYSICS SIMULATOR

**Energy Model.** Formally, SURF-GARDEN embeds a non-Euclidean metric into $\mathbb{R}^3$, inducing negative curvature and curling, an approach grounded in differential geometry and plant morphogenesis research. For a mesh $\mathcal{M}_t = (V_t, E_t, F_t)$, the simulator minimizes a smooth energy Grinspun et al. (2003):

$$\mathcal{E}(V_t) = \underbrace{k_{\text{stretch}} \sum_e (\|e\| - \ell_e^\star)^2 + k_{\text{shear}} \sum_f \|S_f\|_F^2}_{\mathcal{E}_{\text{membrane}}} + \underbrace{k_{\text{bend}} \sum_{(f_i, f_j)} (\theta_{ij} - \theta_{ij}^\star)^2}_{\mathcal{E}_{\text{flexural}}},$$

where $\ell_e^\star$ is the rest length, $S_f$ the shear tensor, and $\theta_{ij}^\star = 0$ the preferred dihedral. The closed-form gradients Tamstorf & Grinspun (2013) are efficient to evaluate and support stable forward Euler steps at $\Delta t = 10^{-2}$.

**Metric Accretion and Non-Euclidean Growth.** Growth is modulated by $g(v) \in [0, 1]$, the normalized geodesic distance to a source set, simulating hormone diffusion (e.g., auxin). Rest lengths update as $\ell_e^\star(t) = ((g(v_i) + g(v_j))/2 + 1)\|e\|$. Edges that exceed $1.5\times$ their original rest length are split. This guided adaptive refinement varies growth spatially in a differential manner, with some parts growing faster than others. Coefficients $k_{\text{stretch}}, k_{\text{shear}}, k_{\text{bend}}$ introduce temporal variation.

**Mesh Quality.** Meshes are optimized with ODT smoothing Chen (2004) and Delaunay edge flips, producing near-isotropic triangles ($0.88 \pm 0.04$ radius-edge ratio) and valence $5-7$. Self-intersections

are avoided using ellipsoidal vertex colliders in a bounding-volume hierarchy (BVH) Wald et al. (2007).

**Topological Variety.** We include six classes of 2-manifolds with boundary: disc, annulus, punctured torus, Möbius strip, pair-of-pants, and thrice-punctured disc. These cover lobed, twisted, and compound forms common in botany and geometry alike, reducing overfitting to select shapes.

**Counterfactual Branching.** Each sequence begins with a 50-frame prefix under baseline parameters $\theta^A$, after which it branches into two or more trajectories with modified controls $\theta^B, \theta^C, \ldots$ (e.g., halved $k_{\text{bend}}$). Identifiers in the half-edge structure ensure that vertex indices and geometry remain aligned post-branch. This enables supervised counterfactual reasoning: identical pasts yield distinct futures, and the model must predict each outcome conditioned on an action token.

Such branch-point supervision equips physically intelligent agents with the ability to forecast the consequences of their own interventions. During training, FOLIAGE observes the prefix through the context encoder and rolls out to each branch using its corresponding action tokens. At inference, alternate futures can be queried by swapping tokens—no simulator calls are required.

SURF-GARDEN provides $7,200$ branched growth sequences, each defined by $k_{\text{stretch}}, k_{\text{shear}}, k_{\text{bend}}$, a topology class, and a random seed. Every sequence spans $400$ frames, evolving from rest to maturity with vertex counts that increase from 20 to $10^5$. Variation is further introduced through quaternion perturbations and vector fields. We split the dataset $8:1:1$ into train/val/test.

### A.1.2 MULTIMODAL CORRESPONDENCE EXTRACTOR AND EVOLUTION TRACING

Each frame yields two mesh-tied modalities. *(1) Multi-view RGB*: eight cameras on a Fibonacci sphere render photorealistic frames with Cycles shading Blender Online Community (2023), HDR lighting, and 50mm lenses. Exposure jitter, defocus, and 20% CutOut masking improve robustness DeVries & Taylor (2017). Each pixel carries its emitting triangle index and barycentric coordinates. Cameras are fixed per trajectory; masks persist over time. *(2) LiDAR-style point cloud*: a $64 \times 2048$ raycast with $\sigma{=}5$ mm Gaussian noise and 5% dropout mimics real-world sparsity. Each point stores its nearest mesh vertex. Both views preserve consistent token indices, enabling direct cross-modal supervision.

Across frames, a half-edge data structure is maintained with a unique identifier for the vertices, edges, and faces. Even as vertices and edges are added (new vertices, edge flips etc.) and their indices are updated, these identifiers remain the same, allowing us to exactly identify the same mesh elements over time alongside their quantities of interest (energy, age, etc.).

### A.2 CRITIC HEADS

For each SURF-BENCH task, we attach a specialized critic to evaluate the core objective under realistic sensor constraints. We denote the learned latent from Foliage as Model-Agnostic Growth Embedding (MAGE). The topology critic ingests mesh-only MAGE into a frozen backbone plus a $768{\rightarrow}256{\rightarrow}6$ MLP for genus classification; the material critic uses a single RGB-based MAGE with a one-hidden-layer regressor to predict bending modulus; the stage critic processes four MAGE embeddings through a 128-unit Bi-GRU Cho et al. (2014) for balanced growth-stage accuracy; the growth critic conditions MeshGPT's Siddiqui et al. (2024) autoregressive split/offset tokens on $M_t$ and $s_{t+\Delta t}$ to measure Chamfer Barrow et al. (1977) and vertex-count errors; the retrieval critic ranks image-to-mesh MAGE by cosine similarity Salton & McGill (1983); and the correspondence critic refines per-vertex features with global and young-region tokens via a 2-layer residual MLP, projects onto 128 spectral components, solves a functional map Ovsjanikov et al. (2012), and applies five ZoomOut refinements Melzi et al. (2019).

### A.3 PERCEPTION STACK DETAILS

At time step $t$, $E_{\text{ctx}}$ ingests multimodal context to produce a compact latent $s_t \in \mathbb{R}^{768}$ which becomes a Modality-Agnostic Growth Embedding (MAGE). An action encoder maps physical control into an action embedding $a_t \in \mathbb{R}^{768}$, which conditions a predictor $P$ to evolve $s_t$ in time to $\hat{s}_{t+\Delta t}$. A target encoder $E_{\text{ctx}}$ augmented with privileged physics features encodes the future world state to $s_{t+\Delta t}$. Critic heads read $(s_t, \hat{s}_{t+\Delta t})$ for downstream tasks.

Each active sensor stream is encoded in a shared token space. This unified representation allows downstream modules to operate purely on token identities, not modalities. This allows missing modalities to be handled gracefully as empty sets for seamless generalization over input combinations.

We distinguish observable inputs—pixels, point coordinates, vertex positions—available at both training and inference, from privileged simulator-only signals: per-vertex stretching and bending energies ($w_{\text{flexural}}, w_{\text{membrane}}$) and material coefficients ($k_{\text{stretch}}, k_{\text{shear}}, k_{\text{bend}}$). The privileged signals influence two paths: gating message passing in AGN, and serving as auxiliary regression targets. The gating path applies a detach, and the auxiliary head is dropped at inference, so no privileged data is needed at test time. All encoders emit tokens in $\mathbb{R}^d$ with $d{=}768$, denoted $\mathcal{T}_I$, $\mathcal{T}_P$, and $\mathcal{T}_M$. Empty inputs yield empty sets, preserving sensor flexibility.

**Image Encoder.** Each RGB frame is resized to $336 \times 336$, partitioned into $16 \times 16$ patches, and fed through a ViT-B/16 Dosovitskiy et al. (2021); the resulting patch embeddings serve as a token set $\mathcal{T}_I = \{\mathbf{p}_k\}_{k=1}^{441} \subset \mathbb{R}^d$

**Point-Cloud Encoder.** Point clouds are encoded by PointNeXt-L Qian et al. (2022) in a two-level PointNet++ Qi et al. (2017) hierarchy, with a final linear projection to $d$. Training augmentations include random point dropout, jitter, and global rotations to mimic LiDAR sparsity. This yields tokens $\mathcal{T}_P = \{\mathbf{q}_k\}_{k=1}^{512} \subset \mathbb{R}^d$

**Accretive Graph Network (AGN).** New mesh vertices emerge during growth; an effective encoder must be invariant to vertex density and sensitive to accretion. Each vertex $v$ gets geometric features $\mathbf{f}_v^{(0)} = [\mathbf{x}_v; \mathbf{n}_v; \kappa_v; b(v)]$. To handle accretion, we introduce Age Positional Encoding (APE): a sinusoidal encoding of vertex birth time $\tau_v \in [0, 1]$ concatenated before diffusion. Then, AGN uses two mesh diffusion layers (DiffusionNet Sharp et al. (2022)) to capture local geometry, followed by two learned-step graph ODEs (GRAND++ Chamberlain et al. (2021); Thorpe et al. (2022)) handles evolving connectivity. This yields the token set $\mathcal{T}_M = \{\mathbf{r}_v\}_{v \in V_t} \subset \mathbb{R}^d$. We found that delaying APE injection degrades performance.

**Geometry-Correspondence Fusion (GCF).** $\mathcal{T}_I$, $\mathcal{T}_P$, and $\mathcal{T}_M$ are synthesized into a unified interaction space via a heterogeneous graph and sparse cross-modal attention. Each token—patch $\mathbf{p}_k$, point $\mathbf{q}_k$, or mesh feature $\mathbf{r}_v$—is a node $i \in V = \mathcal{T}_I \cup \mathcal{T}_P \cup \mathcal{T}_M$, with its $\mathbb{R}^d$ embedding and geometric anchor (barycentric coordinates, 3D position, etc.). Directed edges encode simulator-provided correspondences: $E_{\text{pix}} = \{(\mathbf{p}, \mathbf{r}_v)\}$, $E_{\text{pt}} = \{(\mathbf{q}, \mathbf{r}_v)\}$, $E_{\text{mesh}} = \{(\mathbf{r}_v, \mathbf{r}_u) \mid u \in \mathcal{N}(v)\}$. Learned edge biases $b_{ij}$ encode cross-modal confidence: dot-products for image normals, Gaussian distances for points, and zero for mesh edges. Attention is restricted to edges: $a_{ij} = \mathbf{q}_i^\top \mathbf{k}_j / \sqrt{d} + b_{ij}$, reducing the complexity from $\mathcal{O}(|V|^2)$ to $\mathcal{O}(|E|)$. Tokens communicate via sparse neighborhoods: $\mathbf{u}_i \leftarrow \sum_{j \in \mathcal{N}(i)} \alpha_{ij} \mathbf{v}_j$. We found that GCF layers suffice, since any patch or point is at most two hops from a mesh vertex. Unlike naive concatenation, GCF leverages known correspondences provided by SURF-GARDEN to a complementary, mutually-reinforcing effect: images sharpen mesh features, curvature refines depth, and sparse points gain context.

**Cross-Patch Masking (XPM).** Corrupting the input forces the model to infer missing information from context, driving the encoder to learn robust and semantically rich embeddings rather than relying on trivial correlations. However, generic masking can erase correlated signals or allow for trivial recovery. XPM combats this through three mechanisms: (i) $25\%$ of tokens in each modality are dropped independently, encouraging feature redundancy and stabilizing training. (ii) Paired masking disables neighbors of masked tokens along GCF's correspondence graph edges, blocking trivial copying, and promoting longer-range inference. (iii) A full modality is dropped with $30\%$ probability, sampled after token and pair masking. Training under images-only, points-only, and hybrid conditions promotes invariance and temporal coherence.

**Hierarchical Pooling** first captures local dynamics, then aggregates them into a global summary, so that MAGE reflects both detail and global state. The global token $\mathbf{g}_t = \text{LN}(\frac{1}{|\mathcal{U}_t|} \sum_{u \in \mathcal{U}_t} u)$ aggregates current tokens $\mathcal{U}_t$, remaining invariant to count. The young-region token $\mathbf{y}_t = \frac{1}{|\mathcal{U}_t^{\text{young}}|} \sum_{u \in \mathcal{U}_t^{\text{young}}} \text{LN}(u)$ pools over tokens with age $\tau(u) < 0.2$, capturing fast-changing geometry near new growth. A linear layer with bias $W \in \mathbb{R}^{d \times 2d}$ projects the concatenated pair $(\mathbf{g}_t, \mathbf{y}_t)$ into the final embedding $s_t$. This balances both macroscopic shape and microscopic dynamics.

### A.4 ACTION ENCODER

The action space in our unbounded surface evolution settings consists of the three scalar elastic coefficients that parameterise the shell mechanics, $\mathbf{a}_t^{\text{raw}} = [k_{\text{stretch}}, k_{\text{shear}}, k_{\text{bend}}]$. Because these values span several orders of magnitude, we first apply a logarithmic re–scaling $\tilde{k} = \log_{10}(k)$, $k \in \{k_{\text{stretch}}, k_{\text{shear}}, k_{\text{bend}}\}$ followed by z–score normalization using the mean and variance estimated throughout the training set. The normalized vector $\tilde{\mathbf{a}}_t \in \mathbb{R}^3$ is then embedded into the model's token space through a two–layer perceptron $\mathbf{a}_t = \text{MLP}_{\text{act}}(\tilde{\mathbf{a}}_t), \text{MLP}_{\text{act}} : 3 \to 128 \to d, d = 768$ with GELU activations Hendrycks & Gimpel (2016) and layer normalization Ba et al. (2016). This produces the action token $\mathbf{a}_t \in \mathbb{R}^{768}$).

At training time, the encoder also encounters a learned $\mathbf{a}_{\text{null}}$ embedding that substitutes for $\mathbf{a}_t$ whenever material coefficients are withheld. During training we drop the entire action token with probability 0.1 for this scenario. During inference, the user may supply a physical–coefficient vector to perform counterfactual roll–outs; if omitted, the encoder inserts the null token, reverting the model to passive prediction behavior.

### A.5 ENERGY-GATED MESSAGE-PASSING (EGMP) DETAILS.

Inside $E_{\text{tar}}$ we compute a scalar gate $g_v = 1 + \sigma\big(\text{MLP}(\text{detach}[w_{\text{memb},v}, w_{\text{flex},v}])\big)$ and assemble $G = \text{diag}(g_1, \ldots, g_n)$. The gate modulates the first ODE step $d\mathbf{H} = -G\mathbf{L}\mathbf{H} + G\varphi(\mathbf{H})$, where $\mathbf{L}$ is the Laplacian of the current mesh. We note the choice of a variant robust to open surfaces over the cotangent one Sharp & Crane (2020). High-stress vertices, therefore, propagate messages more rapidly, allowing the latent to focus on regions that are about to wrinkle or curl, while low-stress areas remain stable. In the context branch, the energies are zeroed, so $g_v = 1$ and the update reduces to the standard form. detach keeps the gating weights trainable while treating the energy values as fixed constants, fully preventing the leakage of privileged information.

### A.6 MESH FORECASTING

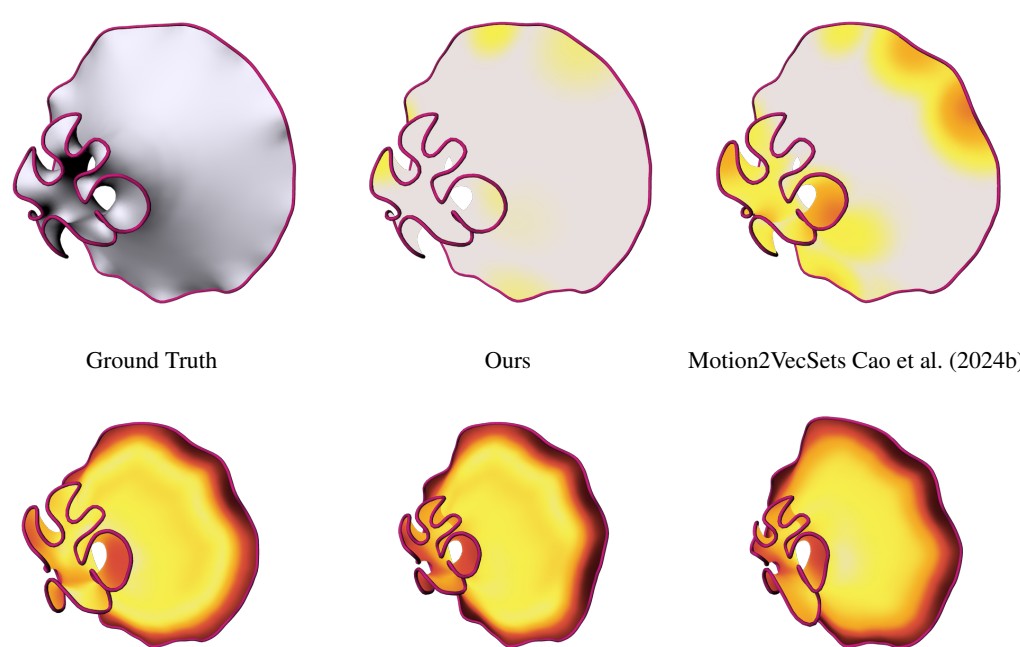

| Ground Truth | Ours | Motion2VecSets Cao et al. (2024b) |
| --- | --- | --- |

| MeshGPT-solo Siddiqui et al. (2024) | CaDeX Lei & Daniilidis (2022) | MeshGraphNets Pfaff et al. (2021) |

Figure 8: Mesh predictions on SURF-BENCH. $\Delta t = 8$, null action.

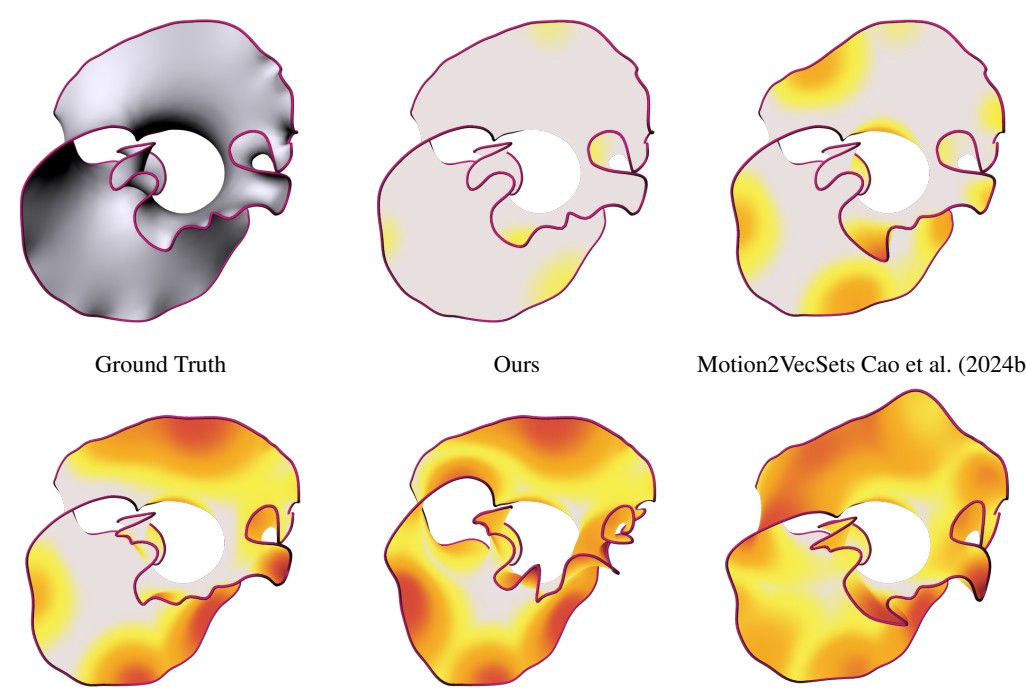

| Ground Truth | Ours | Motion2VecSets Cao et al. (2024b) |

MeshGPT-solo Siddiqui et al. (2024)    CaDeX Lei & Daniilidis (2022)    MeshGraphNets Pfaff et al. (2021)

Figure 9: Mesh predictions on SURF-BENCH. $\Delta t = 10$, action $= [0.05, 0.2, 0.12]$.

Fig. 7, 8, and 9 shows examples of mesh prediction by FOLIAGE and baselines across different look-ahead and action conditioning. For simple topology and limited growth, the general morphology of the surface is preserved. But near the boundaries and in areas of high feature activity (emergence or disappearing of buckling), prediction error increases, especially for baseline models. In Fig. 8, baselines such as MeshGraphNets Pfaff et al. (2021) and CaDeX Lei & Daniilidis (2022) struggle to model surfaces that had enlarged substantially through accretive growth, resulting in visibly 'shrunk-down' predictions. With more complex topology such as the Möbius strip (Fig. 9), these errors propagate globally. These disparities highlight the effectiveness of FOLIAGE's perception-action setup and physics-guided learning to model the complex deformations and growth of the surfaces.

## A.7 CROSS-MODEL RETRIEVAL

In Fig. 10, 11, 12, and 13 we show examples of cross-model retrieval in normal and zero-shot settings for point clouds and images. The correct option (solid line border) is differentiated from the incorrect ones (dashed line border). We note that the purple lines highlighting the boundary (e.g. in Fig. 7) are a visual aid; they are not present in the rendered images of the mesh surface. FOLIAGE's first choice is predominantly the correct one followed by visually similar surfaces (image rendering or point cloud representations) with the same topology categorization. This is observed for unseen examples with complex morphology and challenging viewing angles, indicating the strong semantic awareness and consistency of FOLIAGE's Modality-Agnostic Growth Embedding across different modalities.

## A.8 DENSE CORRESPONDENCE

In contrast to well-studied domains like human or animal bodies, which follow a set template (the skeleton) movements constrained to specific parts of the geometry (e.g. arm movements has very limited impact on the full body), SURF-GARDEN's open surfaces represent a continuum in which features smoothly emerge and dissipate. As Fig. 14 suggests, identifying one of a fixed number of extrusions (e.g. fingers on a hand) is insufficient in the accretive growth regime.

Fig. 15, 16, and 17 show the correspondence maps from a source mesh to a target mesh generated by FOLIAGE and baselines from two distinct viewing angles each. As the surface expands and

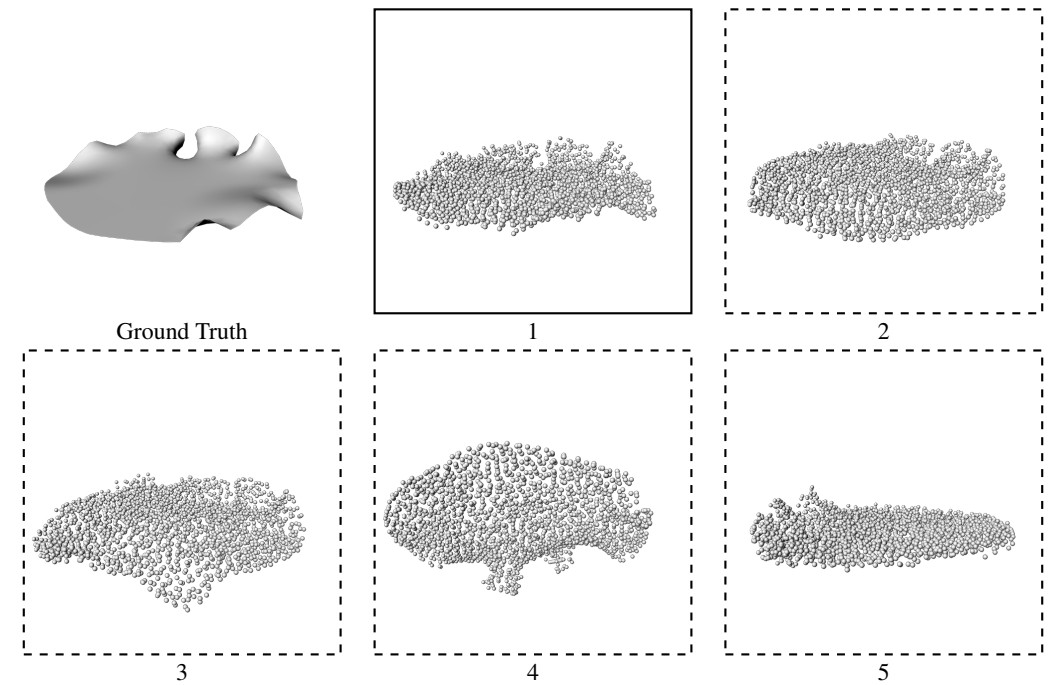

Figure 10: Top-5 retrievals on SURF-BENCH (Image → Point Cloud)

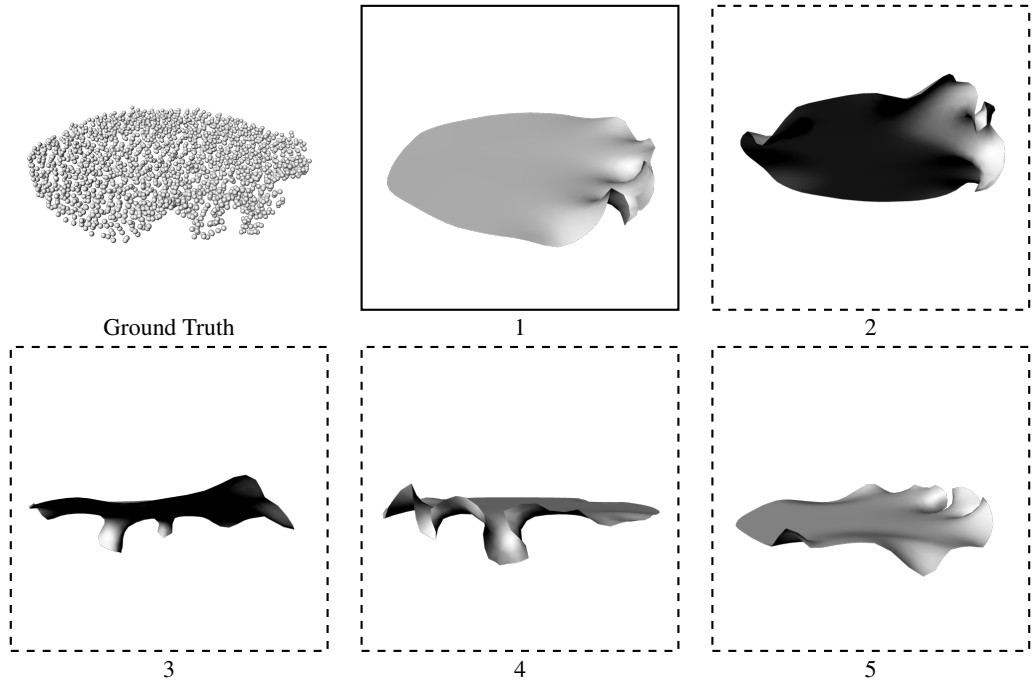

Figure 11: Top-5 retrievals on SURF-BENCH (Point Cloud → Image)

buckles, a small bulge quickly develops into multiple twists and turns which FOLIAGE reliably tracks. Meanwhile, baseline models increasingly lose track of or mismatches features as the morphological complexity of the surface grows under shell physics and material accretion.

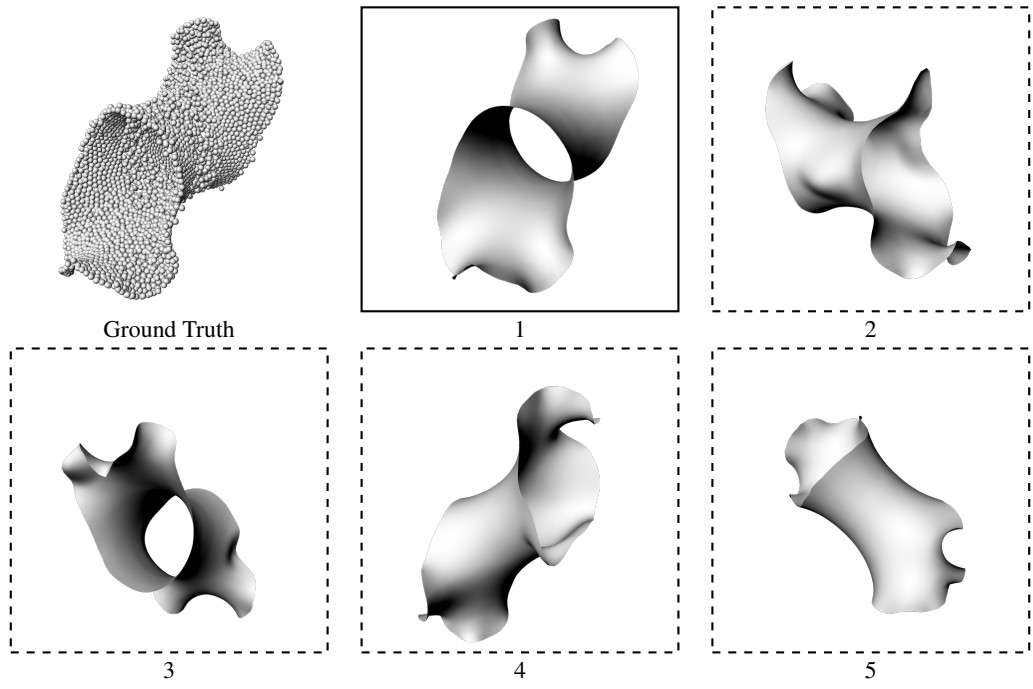

Figure 12: Top-5 retrievals on SURF-BENCH (Point Cloud → Image, Zero-shot)

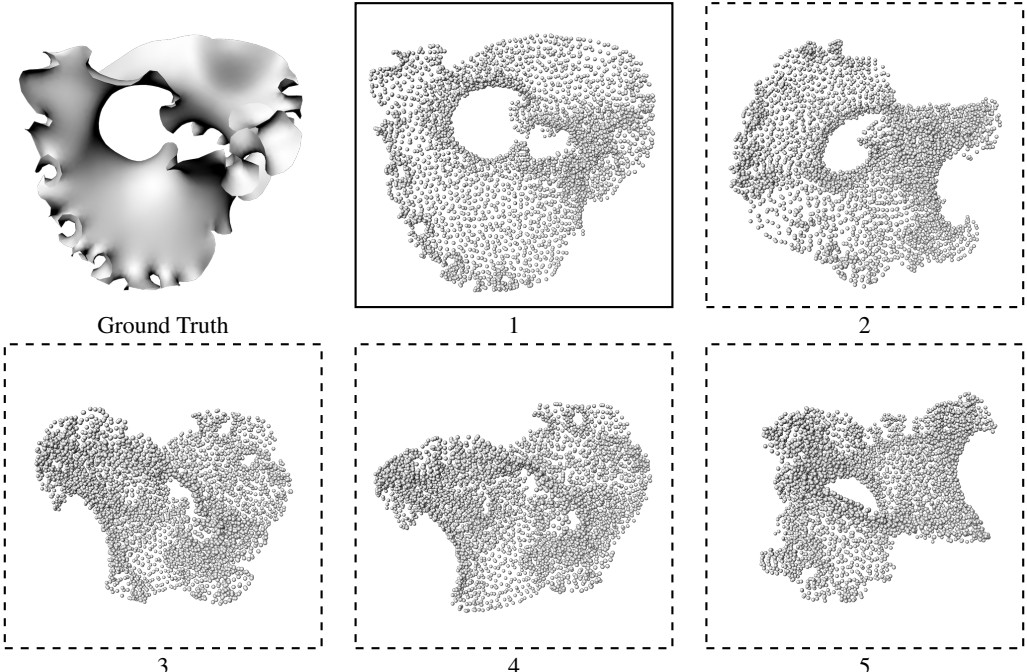

Figure 13: Top-5 retrievals on SURF-BENCH (Image → Point Cloud, Zero-shot)

## A.9 SURF-GARDEN PARAMETERS

SURF-GARDEN supports the exploration of a large morphology space guided by physical control parameters $k_{\text{stretch}}, k_{\text{shear}}, k_{\text{bend}}$. Fig. 18 illustrates the effect of their different combinations. A low bending coefficient models a thinner, more flexible surface that is prone to more complex deformations; a higher value models a thicker surface that only permits large-scale deformations to

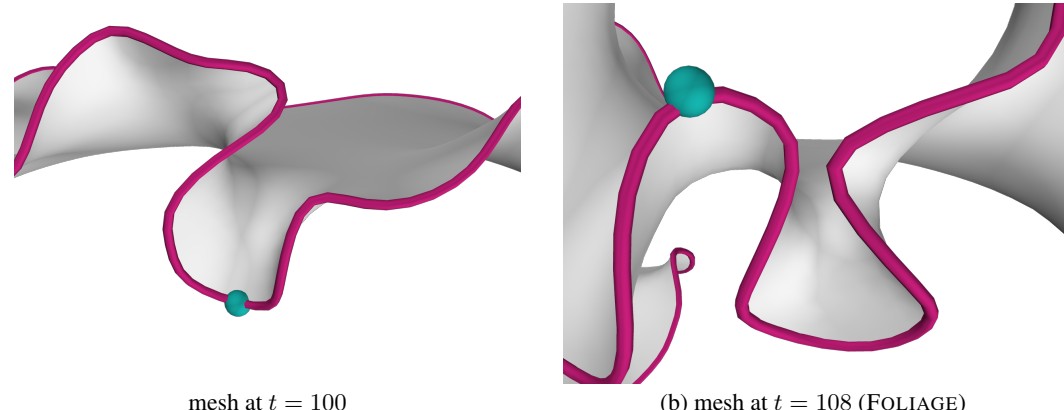

mesh at $t = 100$                    (b) mesh at $t = 108$ (FOLIAGE)

Figure 14: A vertex feature (turquoise sphere) that began at the bottom of a valley (left) quickly develops into the peak of a mountain (right).

form slowly. Stretching and shearing coefficients further regulate the local behavior of the surface, leading to varied morphology.

## A.10 LATENT SPACE

**Topology.** FOLIAGE 's latent space naturally arranges shapes according to their global invariants—genus and boundary count—while not explicitly trained on topology classification tasks. In Fig. 19, all genus-0 surfaces (disc, annulus, pair-of-pants, thrice-punctured disc) form one region, with boundary-count differences causing small shifts along a shared axis: for example, the disc (one boundary) sits between the annulus (two holes) and the pair-of-pants (three holes). By contrast, the genus-1 punctured torus and the non-orientable Möbius strip form a distinct cluster, reflecting their additional "handle" or "twist."

**Forecasting.** We compare two forecasting modes on the SURF-BENCH test set in Fig.20: a direct multi-step predictor that always resets to the true embedding before forecasting, and an autoregressive latent rollout that feeds each prediction back into the model. The solid curve shows that FOLIAGE 's one-shot predictions grow only modestly from approximately 0.03 cm at $\Delta t = 1$ to 0.05 cm at $\Delta t = 8$, demonstrating that its redictor generalizes well beyond its training horizons. The dashed blue curve, by contrast, exhibits a clear "knee" at $\Delta t \approx 4$—early errors accumulate slowly but then accelerate once predictions exceed the $\Delta t \leq 8$ range.

We further plot rollout errors for four prior mesh-prediction methods. MeshGraphNets Pfaff et al. (2021) falters early as more and more vertices are added to expand the surface; CaDeX Lei & Daniilidis (2022) smooths away fine curls in the absence of explicit physics signals; MeshGPT-solo Siddiqui et al. (2024) introduces occasional "ghost" splits under long-range dependency strain; and Motion2VecSets Cao et al. (2024b) blurs high-frequency folds without age-encoding or energy gating. In all cases, these baselines start at higher one-step Chamfer and diverge far more rapidly than FOLIAGE, highlighting the importance of dynamic remeshing, membrane and flexural energy guidance, and robust masking in achieving stable multi-step accuracy.

## A.11 EXTENDED ABLATION STUDIES

Before delving into detailed ablations, we clarify our composite-metric proxy. Rather than tuning four hyperparameters across six individual tasks (and four stress tests), we normalize each task's evaluation metric into a $[0, 1]$ range (inverting distances so that higher score indicates better performance), weight all tasks equally, and sum them into a single scalar. This composite score strongly correlates with the full multi-task performance of interest ($\rho \approx 0.92$), enabling broad five-point sweeps to be run efficiently. Once top-performing settings emerge, we re-evaluate the individual metrics for each task and report them in Tab. 5, 6, and 7.

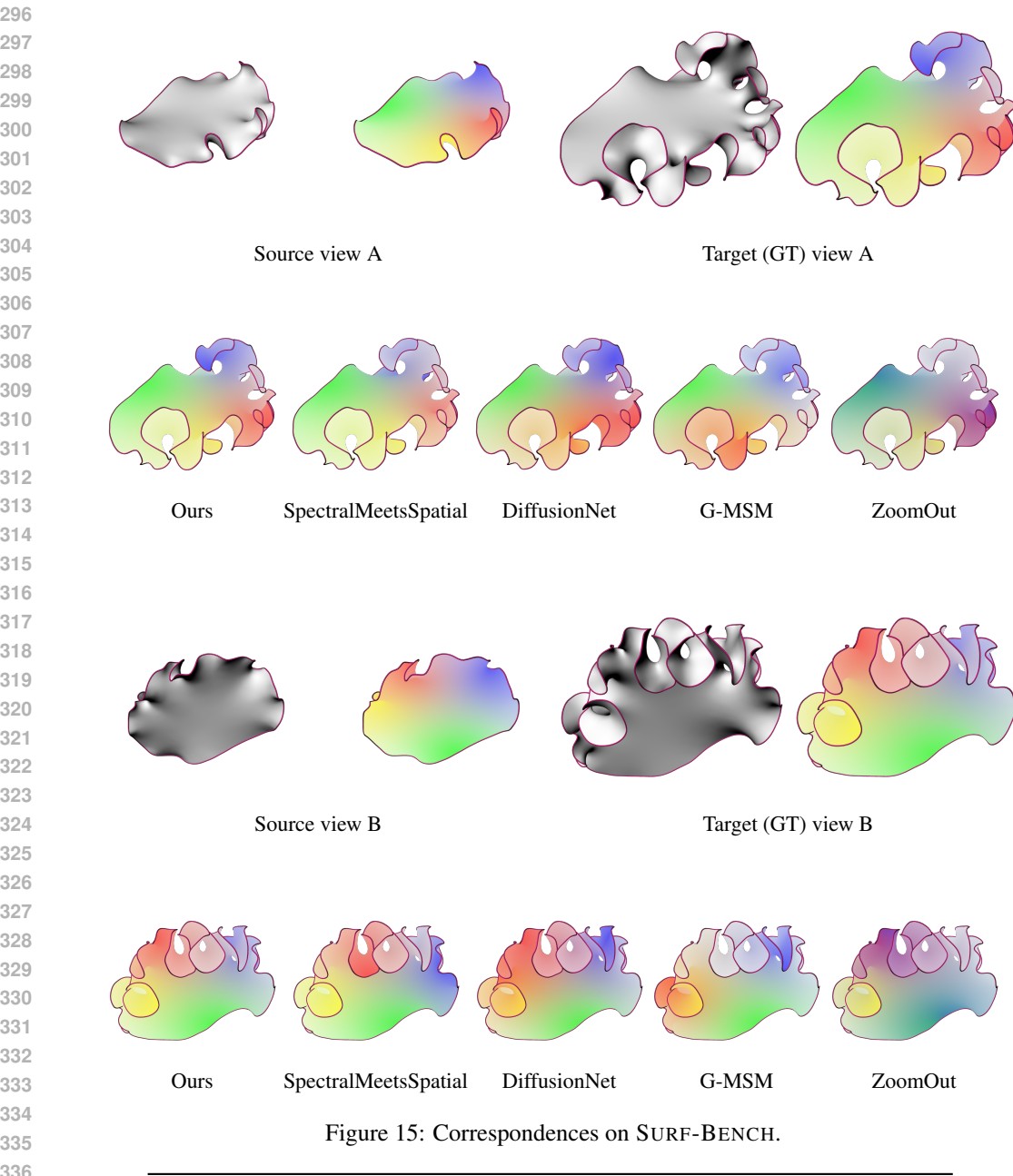

Figure 15: Correspondences on SURF-BENCH.

| Latent Dim. $d$ | Score | EMA Rate | Score | Sampling Range ($\Delta t$) | Score |
|---|---|---|---|---|---|
| 512 | 0.74 | 0.995 | 0.80 | Uniform 1–4 | 0.78 |
| 640 | 0.79 | 0.997 | 0.81 | Uniform 1–6 | 0.80 |
| 768 (Ours) | **0.82** | 0.998 (Ours) | **0.82** | Uniform 1–8 (Ours) | **0.82** |
| 896 | 0.81 | 0.999 | 0.81 | Uniform 1–10 | 0.81 |
| 1024 | 0.78 | 0.9995 | 0.79 | Uniform 1–12 | 0.79 |

Table 5: Ablation results for model capacity and temporal encoding. Each block shows the effect of sweeping a single hyperparameter on the composite validation score.

## A.12 MODEL CAPACITY AND TEMPORAL ENCODING

In Tab. 5, we swept the latent dimensionality $d$ from 512 to 1024. Smaller dimensions (512–640) consistently underperform: the model lacks sufficient capacity to encode fine-grained geometric and

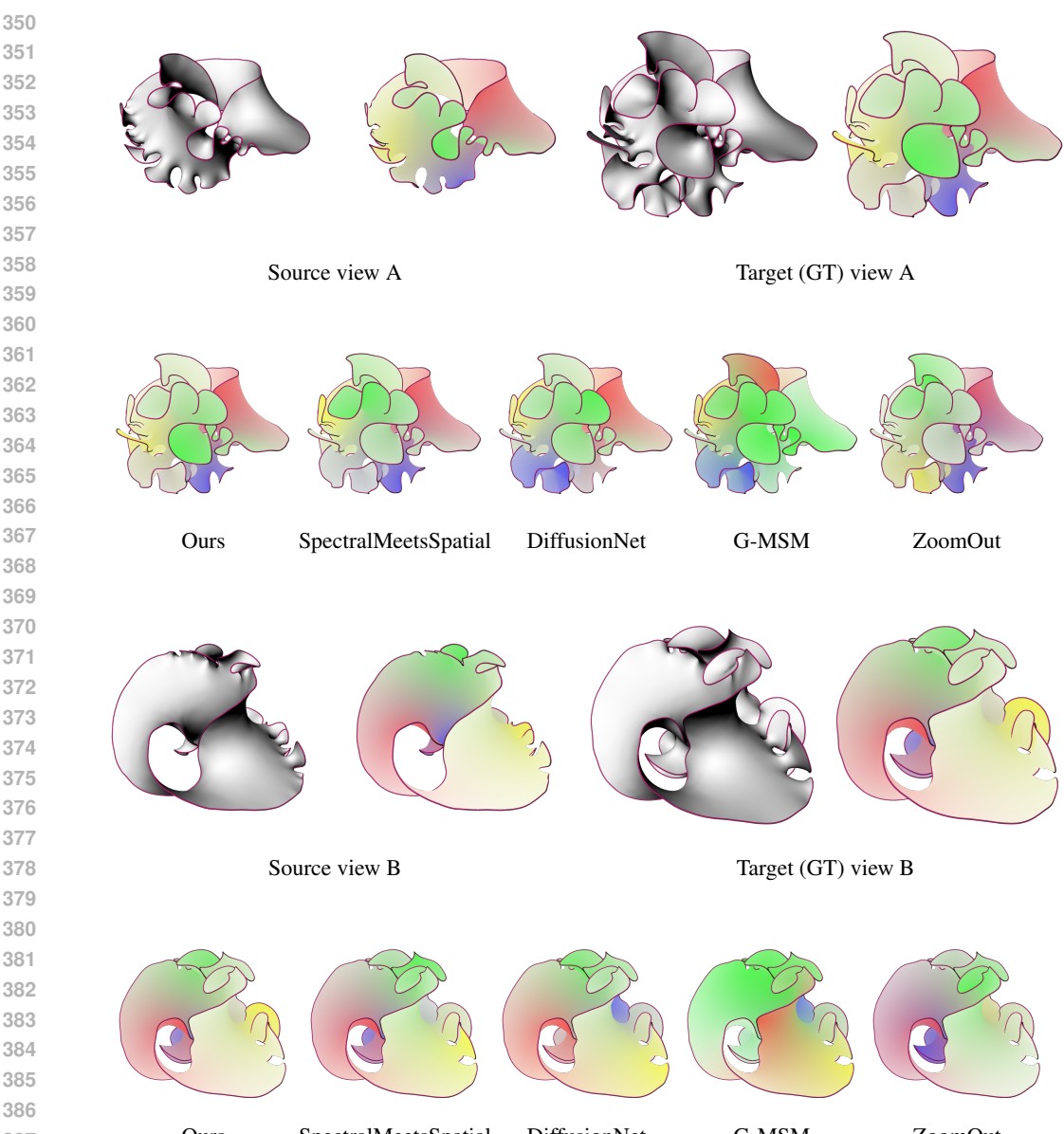

Figure 16: Correspondences on SURF-BENCH.

energetic signals, impairing tasks such as dense correspondence and material regression. Larger dimensions (896–1024) offer diminishing returns—more parameters than data—and exhibit slightly reduced stability during long-horizon rollouts, as the predictor transformer struggles to regularize across a wider channel space. We find $d = 768$ to be the optimal trade-off, balancing expressivity for physics-informed features (e.g., membrane and flexural energies) with trainability.

We also tuned the EMA (exponential moving average) update rate for the target (privileged-signal) encoder. Slower rates (0.995–0.997) update too sluggishly, causing the context and target embeddings to drift apart, which diminishes the effectiveness of energy-gated message passing. Faster rates (0.999–0.9995) over-smooth the target, preventing it from reflecting the latest context weights, and thereby hamper auxiliary energy regression. We observed that a rate of 0.998 best balances stability and responsiveness.

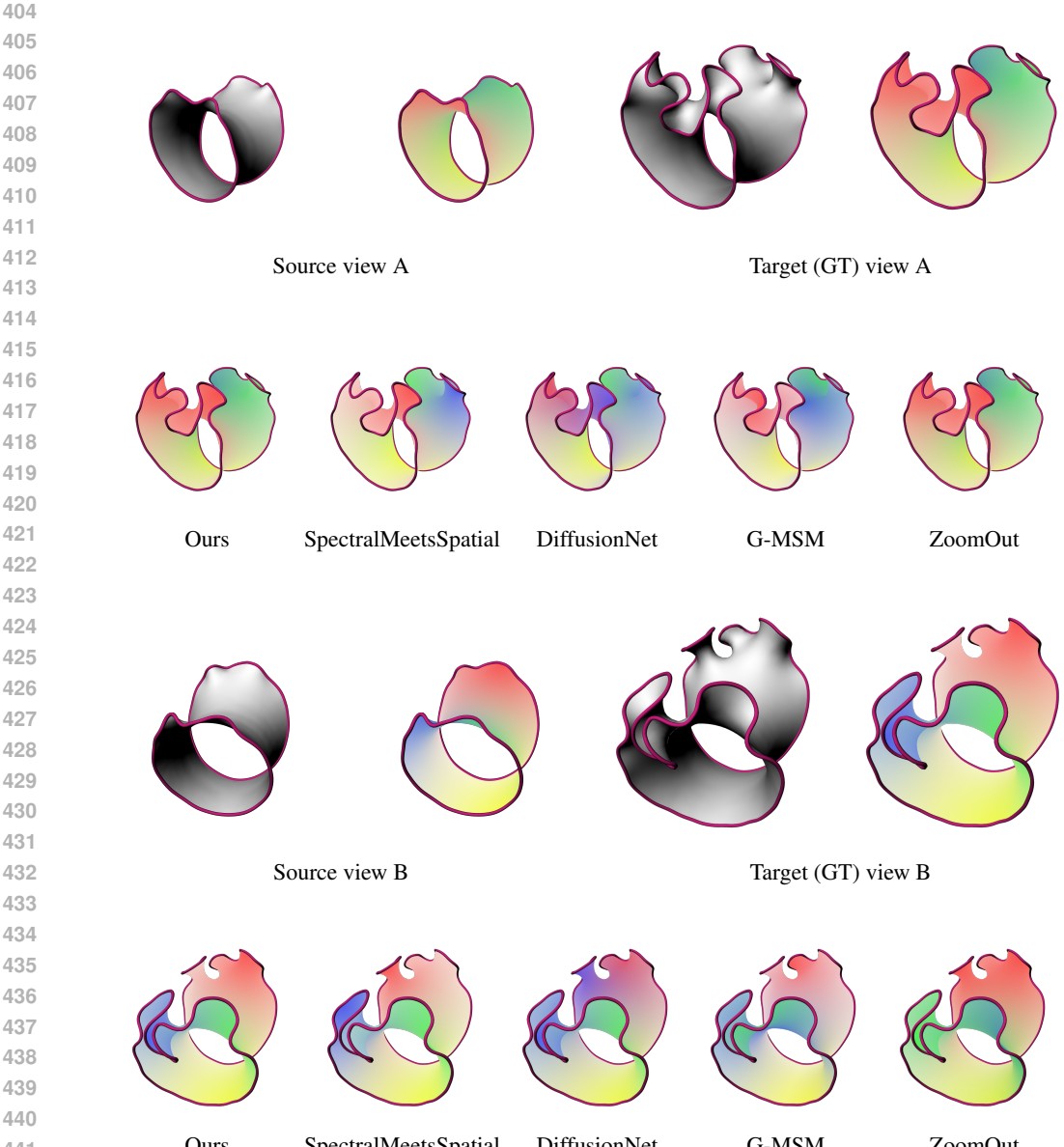

Figure 17: Correspondences on SURF-BENCH.

| Token Drop Ratio | Score | Modality Drop Ratio | Score | Action Drop Prob. | Score |
|---|---|---|---|---|---|
| 15% | 0.78 | 20% | 0.79 | 0% | 0.80 |
| 20% | 0.80 | 25% | 0.81 | 5% | 0.81 |
| 25% (Ours) | **0.82** | 30% (Ours) | **0.82** | 10% (Ours) | **0.82** |
| 30% | 0.80 | 35% | 0.79 | 15% | 0.81 |
| 35% | 0.77 | 40% | 0.75 | 20% | 0.78 |

Table 6: Ablation results for regularization strategies. Each column group shows the effect of sweeping one dropout-related hyperparameter on the composite validation score. The selected configuration for each is highlighted in bold.

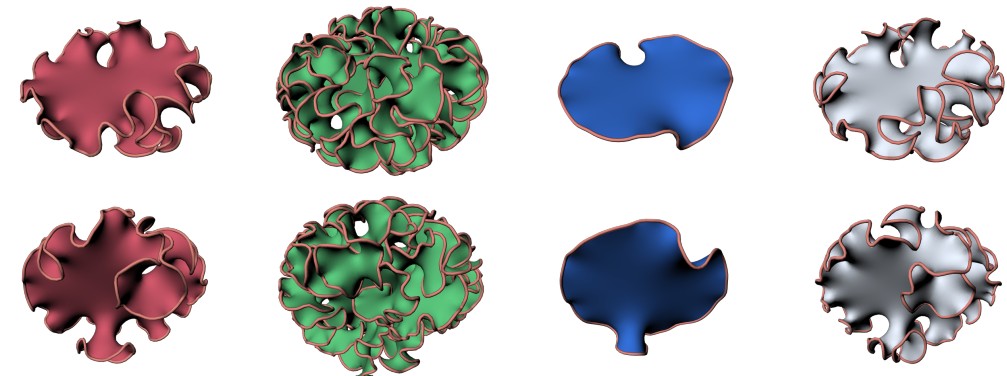

Figure 18: Effect of SURF-GARDEN physical control parameters $[k_{\text{stretch}}, k_{\text{shear}}, k_{\text{bend}}]$ (left to right columns): $[0.15, 0.15, 0.25]$, $[0.15, 0.15, 0.2]$, $[0.1, 0.15, 0.2]$, and $[0.15, 0.1, 0.2]$, respectively.

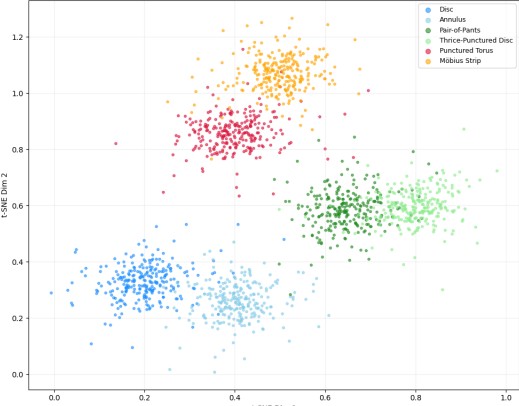

Figure 19: TSNE on topology classes.

A.13 TEMPORAL HORIZON SAMPLING

Choosing the rollout-horizon distribution is critical in a physics-aware world model. As Tab. 5 shows, if we bias sampling toward short horizons (e.g., Uniform(1, 4)), the model learns only incremental dynamics and performs poorly in mid-term predictions; Chamfer and vertex-drift errors spike after 10 steps. Conversely, sampling very long horizons (e.g., Uniform(1, 12)) spreads the model's capacity across a wide temporal range, weakening both short-term fidelity and long-term coherence. Our Uniform(1, 8) policy emphasizes the early and mid-growth phases—where prediction is most critical—while still exposing the model to challenging, longer-range rollouts. This sampling regime consistently maximizes the composite score without overfitting to either extreme.

A.14 REGULARIZATION AND ROBUSTNESS

**Token and Modality Dropout.** In our cross-modal fusion setup, we independently drop 25% of tokens per modality and 30% of entire modalities. As Tb. 6 shows, lower dropout rates (15–20% token, 20–25% modality) fail to regularize adequately: the model overfits to specific sensor patterns and degrades under simulated sensor dropout (Stress S1). Higher rates (30–35% token, 35–40% modality) deprive the fusion transformer of coherent correspondence signals, weakening geometry–correspondence alignment and degrading performance on tasks such as topology classification and retrieval. The selected dropout rates strike a balance, simulating realistic sensor failures without removing so much information that cross-modal attention cannot reconstruct object structure.

**Action Dropout.** We further experimented with dropping the action token during training (0%–20%) in Tab. 6. Omitting action dropout leads to a model that is overly dependent on control inputs and generalizes poorly when such inputs are noisy or absent. Conversely, excessive dropout (15–20%)

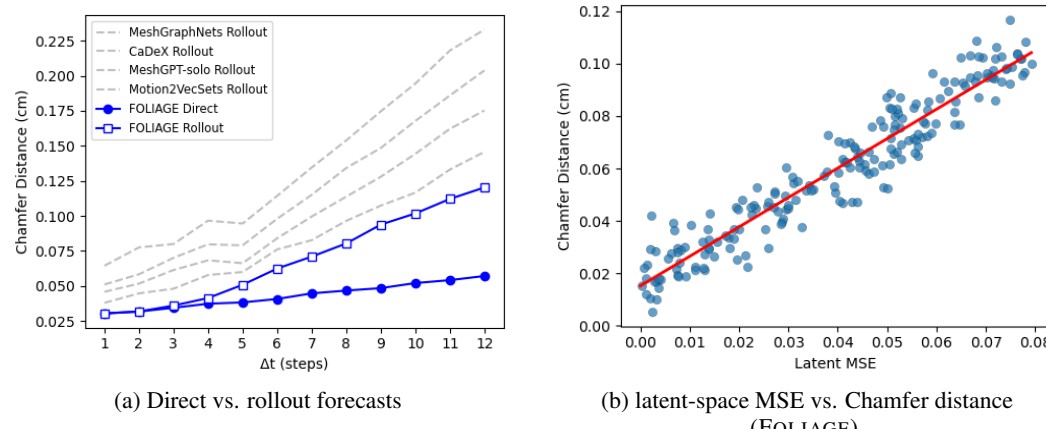

(a) Direct vs. rollout forecasts

(b) latent-space MSE vs. Chamfer distance (FOLIAGE)

Figure 20: Prediction fidelity both in shape space (left) and in its internal latent representation (right) highlighting the effectiveness of FOLIAGE's latent modeling approach for stable long-term rollouts.

| Learning Rate | Score | Weight Decay | Score | $\lambda_E$ | Score | $\lambda_{vc}$ | Score |
|---|---|---|---|---|---|---|---|
| $5.0 \times 10^{-4}$ | 0.78 | $5.0 \times 10^{-3}$ | 0.79 | 0.00 | 0.75 | 0.00 | 0.76 |
| $7.5 \times 10^{-4}$ | 0.80 | $7.5 \times 10^{-3}$ | 0.81 | 0.01 | 0.79 | 0.02 | 0.80 |
| $1.0 \times 10^{-3}$ (Ours) | **0.82** | $1.0 \times 10^{-2}$ (Ours) | **0.82** | 0.02 (Ours) | **0.82** | 0.04 (Ours) | **0.82** |
| $1.5 \times 10^{-3}$ | 0.79 | $1.5 \times 10^{-2}$ | 0.80 | 0.04 | 0.80 | 0.06 | 0.81 |
| $2.0 \times 10^{-3}$ | 0.75 | $2.0 \times 10^{-2}$ | 0.76 | 0.08 | 0.76 | 0.08 | 0.77 |

Table 7: Ablation results for optimization and loss weighting. Each group shows a sweep over one hyperparameter and its effect on the composite validation score.

enforces robustness at the cost of physical consistency, as the model may ignore legitimate control signals. A moderate 10% dropout encourages the model to infer actions from observed state transitions, while still forming tight action–perception loops when control signals are present.

## A.15 OPTIMIZATION

**Learning Rate and Weight Decay.** As Tab. 7 shows, a low AdamW learning rate (e.g., $5 \times 10^{-4}$) leads to slow convergence and under-optimized parameters, while a high learning rate (e.g., $2 \times 10^{-3}$) causes unstable gradients, especially in the multi-head self-attention layers of the predictor. Similarly, a weak weight decay (e.g., $5 \times 10^{-3}$) under-regularizes the high-dimensional latent space, whereas overly strong decay (e.g., $2 \times 10^{-2}$) suppresses meaningful emergent physics representations. Our chosen configuration—learning rate of $1 \times 10^{-3}$ and weight decay of $1 \times 10^{-2}$—yields smooth optimization and robust generalization.

**Energy and Variance–Covariance Loss Weights.** The auxiliary energy regression weight $\lambda_E$ and variance–covariance regularizer $\lambda_{vc}$ govern how much the model prioritizes privileged physical signals over raw rollout accuracy. Setting the energy weight to zero ($\lambda_{\mathcal{E}} = 0$) causes material inference to degrade, while excessive weight (e.g., $\lambda_E = 0.08$) pulls the latent space toward physics features at the cost of open-loop prediction accuracy, worsening Chamfer and drift metrics. We select $\lambda_E = 0.02$ and $\lambda_{vc} = 0.04$ to ensure that physics cues meaningfully inform the representation without overwhelming the primary learning signal, striking a balance between interpretability and predictive performance.

