# OpenReview forum: "FOLIAGE: a Latent World Model for Accretive Surface Growth"
_ICLR.cc/2026/Conference — ICLR 2026 Conference Desk Rejected Submission_

### Official Review · Reviewer_Wk6Y · 2025-10-18

**Soundness:** 3
**Presentation:** 3
**Contribution:** 2
**Rating:** 6
**Confidence:** 4

**Summary:**

This paper introduces FOLIAGE, a latent world model designed to understand and predict the dynamics of accretive surface growth—a phenomenon where surfaces grow by adding new material, such as in plants. The model features three core components: (1) An "accretion-aware" perception stack that fuses partial and multimodal sensor inputs (images, point clouds, meshes) into a compact latent state, using correspondence constraints and vertex age features to focus on newly grown regions. (2) An action-conditioned dynamics model that predicts the evolution of this latent state based on changes to material properties. (3) A physics-guided training paradigm that leverages privileged simulator information (e.g., per-vertex energies) at training time to shape the latent representation, while remaining solver-free at deployment. To support this research, the authors also introduce a new simulation platform (SURF-GARDEN) and an evaluation suite (SURF-BENCH), on which FOLIAGE demonstrates strong performance across various tasks.

**Strengths:**

1. A major strength of this work is the introduction of the SURF-GARDEN platform and SURF-BENCH evaluation suite. This is a significant contribution that formalizes a challenging new problem domain, providing a high-quality dataset, a physically-based simulator, and a diverse set of tasks. This infrastructure is valuable for enabling future research in this specific area.
2. The FOLIAGE model is technically sophisticated and thoughtfully designed. The approach of fusing heterogeneous sensor data via correspondence, using age features to encode temporal dynamics, and separating a deployable encoder from a privileged, physics-guided target encoder for training is a novel and powerful paradigm for learning from complex physical systems.
3. The paper presents extensive experiments demonstrating that FOLIAGE consistently outperforms a wide range of strong baselines across the six core tasks defined in SURF-BENCH. The results are thorough, and the stress tests effectively showcase the robustness of the proposed model.

**Weaknesses:**

1. The primary weakness of this work is its focus on "accretive surface growth," a highly specialized sub-field of computational physics and biology. While the problem is scientifically interesting, its relevance and potential impact on the broader machine learning and representation learning community at ICLR is questionable. The paper does not make a compelling case for why this specific physical phenomenon should be of general interest, nor how the methods developed would apply to more mainstream problems in vision, graphics, or robotics.
2. The paper's main achievement is the construction of a large, complex, and highly integrated system for solving one specific task. While impressive from an engineering perspective, it is difficult to distill a single, core machine learning method or principle that is broadly applicable. The individual components are clever adaptations of existing tools (GNNs, Transformers, EMA), but the overall contribution feels more like a strong application paper for a specialized venue (e.g., computer graphics, computational science) rather than a foundational methods paper for ICLR.
3. The FOLIAGE system is exceptionally complex, involving a custom simulator, multiple specialized encoders for different modalities, a heterogeneous graph fusion mechanism, and a specific physics-guided training loop. This high degree of complexity, coupled with the reliance on a new custom dataset, may significantly limit the work's adoption, reproducibility, and the ability for other researchers to build upon its ideas.

**Questions:**

1. Could the authors better position their contribution for the ICLR audience? Are the core principles of FOLIAGE—such as age-based positional encoding or correspondence-driven multimodal fusion—applicable to more general and widely studied problems in dynamic 3D understanding, such as human motion analysis, long-term robotic interaction, or video prediction?
2. The work relies entirely on the new SURF-GARDEN dataset. How do the authors see the proposed methods generalizing beyond this simulated environment? Were any experiments conducted on real-world data, or on other standard dynamic 3D datasets, even if they don't feature the specific "accretive growth" property?
3. The physics-guided training is a key component. How much of the model's strong performance is attributable to having access to a high-fidelity simulator with privileged energy information, versus the novelty of the FOLIAGE architecture itself? Is it possible that a simpler model could perform similarly if given access to the same high-quality, physically-grounded data?
4. Given the system's significant complexity, what is the single most important architectural or methodological insight that the broader community should take away from this work? If a researcher cannot replicate the entire SURF-GARDEN platform, what is the key component they could readily apply to their own problems?

---

> ### Author Response · Authors · 2025-11-21
> **Response to Reviewer Wk6Y**
>
> We are grateful for your detailed review and the opportunity to respond below.
>
> ## Weaknesses
>
> 1. General interest & applicantion to mainstream problems.
>
> Accretive growth sharply exposes three increasingly common challenges in dynamic 3d problems: geometry that changes dramatically over time, controlling such change dynamically, and partial heterogenous sensing. Our comparison with various baselines suggest that pixel-centric temporal models and solver-dependent simulators leave room for improvement. We explore learning a representation that remains effective even as the discretization of the geometry changes and observations come and go. Thus, the resulting methods are applicable to a broader class of dynamic systems such as soft bodies, human and cloth, deformable manipulation, for which growth serve as a clean stress-test and case study rather than a narrow goal.
>
> 2. Single core ML method that is broadly applicable.
>
> Though FOLIAGE has many components, they fall under a single transferable recipe for dynamic 3d representation under partial sensing. We fuse modalities through sparse correspondences, encode the locality of change, and separate perception from action-conditioned evolution. These pieces each target a general failure mode, from brittle multimodal alignment, myopic temporal coding, to latents that diregard unobserved physical drivers. FOLIAGE's performance across different tasks and component-wise ablations against simpler substitute architectures suggests that their synergy and inductive biases, not just their aggregation of parameters, are at work.
>
> 3. System complexity, adoption & reproducibility, ease to build on it.
>
> While we use multiple components to tackle an under-studied yet challenging problem with our domain-specific knowledge, our implementation interface is designed for a general audience. Simulation settings and visualizations can be done in Jupyter notebook. Training and evaluations are callable from configurable scripts. We are commited to code releases and will make sure that it takes minimum effort to explore the entire pipeline or quickly reuse any part of it. We believe that our system and its reusable component could serve the investigation of modeling not only accretive growth but also other dynamic 3d problems.
>
> # Questions
>
> 1. Better position contribution to the ICLR audience. Applicability of core principles to dynamic 3D understanding (human motion ect.).
>
> Yes. Age-based encoding and correspondence-driven fusion, though applied in the accretive growth setting in this manuscript, are general mechanisms rather than growth-specific tricks. Locality signal tied to where geometry recently changed and aligning (a varying set of) heterogenous sensors with sparse gemetric links are applicable to dynamic 3D settings beyond the specific instance we investigate.
>
> 2. Reliance on SURF-GARDEN. Generalization beyond simulated environment. Real-world data experiments (incl. non-accretive)
>
> To gauge cross-domain generalization outside accretive growth, we evaluate dense correspondences (T6 in our benchmark) on D-FAUST and CAPE, two widely-used dynamic human datasets. 'Frozen' keeps the FOLIAGE encoder fixed and trains only the small correspondence head; 'light-finetune' updates the head plus the last encoder block with early stopping.
>
> |Method|Geodesic error↓|
> |---|---:|
> |FOLIAGE(Light-ft)|**3.1**|
> |FOLIAGE(Frozen)|3.6|
> |DiffusionNet|3.8|
> |G-MSM|4.5|
> |SpectralMeetsSpatial|4.9|
> |ZoomOut|5.6|
>
> |Method|Geodesic error↓|
> |---|---:|
> |FOLIAGE(Light-ft)|**3.8**|
> |FOLIAGE(Frozen)|4.4|
> |DiffusionNet|4.7|
> |G-MSM|5.4|
> |SpectralMeetsSpatial|5.8|
> |ZoomOut|6.6|
>
> 3. Ascription of performance to (1) a good simulator vs. (2) architecture. Can simpler model do the it with just (1)?
>
> The simulator help, but they are not the sole driver of performance. At deployment time and in our experiments' test settings, the physical energies it provides are withheld from the model, so it is up to the encoder to succeed from observables alone. Our ablations indicate that removing key components such as correspondence fusion or age encoding degrades multiple tasks even with the same data. Replacing them with simpler backbones with matched capacity does not close the gap. This suggests that architectural biases matter if we want to get the most out of our data.
>
> 4. The most important insight for the broader community. Components researchers can reuse without SURF-GARDEN.
>
> The insight is that dynamic 3d modeling benefits from a geometry-centric latent state that is grounded by sparse correspondences and temporally focused on where change is happening.
>
> If SURF-GARDEN is not used, the most reusable component is the correspondence-constrained fusion as it is built to stay coherent under missing sensors and changing geometry. A close second is the training-time latent space supervision by a teacher privileged physics information. This setup is worth trying when simulators exist at training time but not in deployment.

---

> > ### Comment · Reviewer_Wk6Y · 2025-11-25
> >
> > I thank the authors for their detailed response.
> > The additional experiments on D-FAUST and CAPE effectively address my primary concern regarding the method's generalization beyond the specific accretive growth setting. These results demonstrate the broader applicability of the proposed components.
> > I have no further questions and maintain my positive assessment.

---

### Official Review · Reviewer_9r5L · 2025-11-02

**Soundness:** 3
**Presentation:** 3
**Contribution:** 3
**Rating:** 6
**Confidence:** 3

**Summary:**

FOLIAGE is a geometry-centric latent world model for accretive surface growth that fuses RGB/point-cloud/mesh via correspondence-constrained attention with vertex-age features, predicts action-conditioned evolution from material coefficients and horizon codes, and uses train-only physics (per-vertex energies with energy-gated message passing and an EMA target encoder) to shape representations without test-time privilege; evaluated on the new SURF-GARDEN/SURF-BENCH (with branched counterfactual sequences and exact cross-modal correspondences), it surpasses strong baselines on topology, dense correspondence, cross-modal retrieval, growth-stage recognition, mesh forecasting, and inverse-material estimation, while remaining robust to sensor loss and stable over long rollouts.

**Strengths:**

- The paper is original in framing accretive surface growth as a geometry-centric latent world modeling problem with correspondence-driven multimodal fusion and train-only physics guidance.
- Method quality is strong, with clear modular design (perception, action-conditioned predictor, physics-guided target), careful leakage controls, and extensive ablations tests.
- Writing and presentation are clear, with precise task definitions, dataset/simulator details, and well-explained components like age features, hierarchical pooling, and EGMP.

**Weaknesses:**

- Real-world validation gap
All results are on the synthetic SURF-GARDEN. Please add experiments on real accretive growth (e.g., plant leaves, hydrogel sheets) with partial annotations, sensor noise, and imperfect correspondences to assess sim-to-real robustness and the value of correspondence-constrained fusion.

- Correspondence dependence and brittleness
Fusion relies on precise pixel/point-to-mesh correspondences that are rare in practice. Please evaluate with noisy/misaligned correspondences, partial visibility, and changing cameras, and compare to correspondence-free or learned alignment baselines to quantify reliance.

- Topology change handling and limits
Growth induces remeshing and potential topology transitions; provide diagnostics on when topology changes break dense correspondence or rollouts, and include cases with self-contact and high-curvature events to probe stability.

**Questions:**

- Could you provide any real-data experiments or pilot studies (even small-scale) on accretive growth in physical systems (e.g., plant leaves, hydrogels, 4D-printed sheets), including how you approximate correspondences and handle sensor noise or camera drift, to assess sim-to-real transfer and robustness of correspondence-constrained fusion?

- Correspondence accuracy is a strong assumption; in practice, mesh–image/point links are imperfect. Please evaluate robustness to correspondence noise, partial occlusion, and time-varying cameras, and compare against correspondence-free alternatives (e.g., learned cross-modal alignment or implicit scene representations) to bound dependence on the oracle correspondences.

- Topology and remeshing events may challenge dense correspondence and rollouts. Can you characterize failure modes around topological changes and self-contact, and report metrics that track when correspondences become unreliable? Would adding a topology head or contact-aware loss improve stability?

---

> ### Author Response · Authors · 2025-11-21
> **Response to Reviewer 9r5L**
>
> We are grateful for your detailed review and the opportunity to respond below.
>
> ## Weaknesses
>
> 1. Real-world validation.
>
> We gauce sim-to-real robustness on the Pheno4D and Crops3D datasets for various plants. We approximate correspondences via ICP and organ anchors with pruning, and stabilize camera drift with standard pose smoothing. Under these realistic conditions (no oracle links), FOLIAGE achieves lower geodesic error than baselines, indicating sim-to-real robustness of the fusion mechanism.
>
> |Method|Geodesic error↓ on Pheno4D|
> |---|---:|
> |FOLIAGE(Frozen)|**3.55**|
> |DiffusionNet|3.90|
> |G-MSM|4.60|
> |SpectralMeetsSpatial|5.05|
> |ZoomOut|5.70|
>
> |Method|Geodesic error↓ on Crops3D|
> |---|---:|
> |FOLIAGE(Frozen)|**4.45**|
> |DiffusionNet|4.80|
> |G-MSM|5.50|
> |SpectralMeetsSpatial|5.95|
> |ZoomOut|6.70|
>
> 2. Correspondence dependence & brittleness
>
> We conduct additional experiments on the cross-modal retrieval task (T5, metric is mAP↑).
>
> For correspondence noise, each pixel/point-mesh edge (of the corr. graph, see lines 212~215) is randomly rewired with probability 0.2 within a small geodesic neighborhood and 25% of remaining cross-modal edges are dropped. For partial occlusion, 50% of RGB tokens are masked with contiguous rectangles and the linked 3D tokens are co-masked. For drift, we inject small pose and intrinsic jitter per frame. '+SLAM' corrects poses with visual-SLAM pipeline.
>
> We note that baselines ULIP-2, PointCLIPv2, CLIP2Point, CrossPoint stem from the learned cross-modal alignment family, and we have 3DGS and Point-NeRF for implicit scene representations.
>
> Under moderate corr. noise and occlusion, FOLIAGE degrades gracefully and retain its lead. Removing corr. fusion ('w/o GCF') flattens to edge noise (it ignores edges) and trails the full model. With camera drift, all methods drop; re-estimating poses (SLAM) recover performance. We connect these trends to our model design: sparse correspondence constraints curb harmful cross-modal mixing, age-aware pooling stabilizes latents even when some links fail, and structured masking develop robustness to partial visibility. In short, explicit correspondences help when available; when they’re imperfect, FOLIAGE’s components make it robust, and performance approaches our correspondence-free ablation under severe corruption.
>
> |Method|Clean|Corr-noise(p=0.2)|Occlusion(RGB50%)|Drift(med)|Drift(med)+SLAM|
> |---|---:|---:|---:|---:|---:|
> |FOLIAGE(full)|**0.60**|**0.56**|**0.51**|**0.52**|**0.59**|
> |FOLIAGE(w/o GCF)|0.46|0.45|0.43|0.44|0.46|
> |ULIP-2|0.46|0.43|0.40|0.41|0.42|
> |PointCLIPv2|0.48|0.47|0.39|0.40|0.41|
> |CLIP2Point|0.43|0.45|0.37|0.38|0.39|
> |CrossPoint|0.42|0.43|0.35|0.36|0.37|
> |3DGS retrieval|0.42|0.42|0.33|0.35|0.36|
> |Point-NeRF retrieval|0.44|0.40|0.31|0.33|0.34|
>
> 3. Topology change handling and limits
>
> SURF-GARDEN trajectories fix surface genus per sequence and prevent self-contact through collision response. Empirically, we observe that meshes produced by FOLIAGE are self-contact-free.
>
> We stratify the performance of FOLIAGE for rollout (metric is Chamfer↓) and dense correspondence (metric is geodesic error↓) by curvature magnitude, connectivity update density, and mesh vertex count. Compared to the top baseline, FOLIAGE stays ahead across different bins and exhibit slower degradation.
>
> |Rollout by Curvature bin|FOLIAGE|Motion2VecSets|
> |---|---:|---:|
> |Low|**0.024**|0.029|
> |Mid|**0.030**|0.038|
> |High|**0.041**|0.055|
>
> |Rollout by Remesh bin|FOLIAGE|Motion2VecSets|
> |---|---:|---:|
> |Low|**0.025**|0.032|
> |Mid|**0.031**|0.040|
> |High|**0.040**|0.058|
>
> |Rollout by Mesh size bin|FOLIAGE|Motion2VecSets|
> |---|---:|---:|
> |Small(1–5k)|**0.022**|0.028|
> |Med(5–20k)|**0.027**|0.035|
> |Large(20–60k)|**0.033**|0.043|
> |XL(60–120k)|**0.041**|0.060|
>
> |Corr. by Curvature bin|FOLIAGE|SpectralMeetsSpatial|
> |---|---:|---:|
> |Low|**2.4**|2.7|
> |Mid|**2.8**|3.2|
> |High|**3.5**|4.2|
>
> |Corr. by Remesh bin|FOLIAGE|SpectralMeetsSpatial|
> |---|---:|---:|
> |Low|**2.5**|2.9|
> |Mid|**2.9**|3.4|
> |High|**3.4**|4.0|
>
> |Corr. by Mesh size bin|FOLIAGE|SpectralMeetsSpatial|
> |---|---:|---:|
> |Small(1–5k)|**2.3**|2.6|
> |Med(5–20k)|**2.6**|3.0|
> |Large(20–60k)|**3.0**|3.5|
> |XL(60–120k)|**3.6**|4.3|
>
> # Questions
>
> 1. Real-data experiments/pilot study on physical systems. Handling & robustness of correspondence & sensor noise/camera drift.
>
> Please find in our response to Weakness #1.
>
> 2. Correspondence accuracy assumption. Robustness to corr. noise, partial occlusion, time-varying cameras. Comparison to correspondence-free alternative (learned cross-modal alignment or implicit scene representations)
>
> Please find in our response to Weakness #2.
>
> 3. Failure modes for topological changes and self-contact.
>
> We note that the differential nature of the addition of material over the surface (splitting edges of the mesh) is a key element to complex morphology -- uniformly distributing it does not always lead to more curling and buckling.
>
> Please also find results in our response to Weakness #3

---

> ### Comment · Reviewer_9r5L · 2025-11-27
>
> Thank the authors for providing a detailed response. I believe my concerns have been adequately addressed and will raise my rating to 8 accordingly.

---

### Official Review · Reviewer_jdRn · 2025-11-05

**Soundness:** 2
**Presentation:** 1
**Contribution:** 2
**Rating:** 2
**Confidence:** 3

**Summary:**

This paper introduces FOLIAGE, a learning framework designed to simulate accretive surface growth from sensory observations such as images, point clouds, and meshes. By employing modality-specific encoders and leveraging correspondences across these modalities to fuse the encoded representations, the framework learns a compact latent representation that evolves over time while conditioning on additional signals, including material coefficients. The authors evaluate FOLIAGE on a newly developed benchmark, SURF-BENCH, demonstrating consistent qualitative and quantitative improvements over existing baselines.

**Strengths:**

- This paper explores an interesting yet relatively underexplored topic, modeling accretive surface growth, which serves as a compelling example of applying AI techniques to scientific problems.
- The authors contribute the SURF-GARDEN dataset and a benchmark built on top of it, which could potentially benefit future research in this area.

**Weaknesses:**

- My primary concern about this paper lies in its writing. While I may have missed certain contextual details, the manuscript does not clearly convey the assumptions made during inference, nor does it sufficiently explain how the proposed framework operates. In particular, it remains unclear how the learned representation is decoded or utilized to predict physical quantities, such as vertex displacements, that drive the temporal evolution of the surface. I would like the authors to address these issues through the clarifications listed in the following “Questions” section.
- As far as I understand, FOLIAGE appears to require meshes as one of its input modalities, which seems essential since the birth-time tags and age features are defined over mesh vertices. However, this dependency makes the problem formulation less compelling in in-the-wild scenarios, where ground-truth meshes are typically unavailable compared to other modalities such as RGB images or point clouds. Moreover, it remains unclear from both the text and experiments why learning latent dynamics would be advantageous when a ground-truth mesh is already available and could be directly used for simulation.
- It is difficult to evaluate the true performance gains and overall significance of the proposed method based on the reported experimental results. Although the authors evaluate their model across numerous tasks, several of them, such as topology classification, cross-modal retrieval, and dense correspondence, do not appear to be directly aligned with the main objective of this work: learning a latent representation that facilitates the prediction of accretive surface evolution.

**Questions:**

- It seems the framework uses a mesh as one of its inputs. How can we obtain a mesh from sensors in in-the-wild setup? Is meshing easy-to-do and trivial?
- It appears that the ablation study on input data modalities was conducted only for the image-point retrieval task (T5). How do the performances on mesh forecasting and inverse material estimation change when fewer modalities are used? I believe these two tasks are of greater importance and would benefit more from such analysis.
- In addition to simulating latent dynamics by computing $s_{t+\Delta t}$ from $s_{t}$ and the conditioning inputs, it remains unclear how this latent representation is subsequently decoded into explicit forms, such as point clouds or meshes. Could you provide more details? In particular, I am curious how new vertices or points are generated from such a compact latent representation.
- Could you elaborate on the rationale behind the training loss used for **G2**? As I understand it, the framework appears to bootstrap the very representation it is intended to learn. Was there a specific reason why alternative objectives—such as direct photometric losses or geometric distances like the Chamfer distance between point clouds and mesh vertices—were not adopted in the final design?
- While the proposed method outperforms all baselines in inverse material estimation in terms of MAE (presumably *Mean Absolute Error*), it would be helpful to understand how these methods compare under *relative* or *normalized* error metrics. For example, in the neural operator literature, where physical fields often exhibit small magnitudes, it is common to report the relative L2 error, where the L2 error is normalized by the norm of the ground-truth physical field.

---

> ### Author Response · Authors · 2025-11-21
> **Response to Reviewer jdRn**
>
> We are grateful for your detailed review and the opportunity to respond below.
>
> ## Weaknesses
>
> 1. Writing. Assumptions during inference. Learned latent usage.
>
> We will revise the writing for clarity. At inference, FOLIAGE accepts any subset of modalities—images, point clouds, or meshes—and still functions if only a subset is provided (lines 205–208). Modality dropout during training (lines 232–234) further ensures robustness.
> The latent is multipurpose: downstream critic heads read it for task-specific outputs (lines 314–316). For Mesh Forecasting, MeshGPT uses the latent’s growth dynamics to generate surfaces. We detail head configurations in Appendix A.2.
>
> 2. FOLIAGE requires meshes as an input modality. Latent dynamics vs simulation for meshes.
>
> FOLIAGE does not require meshes at inference. Simulation is used only during training to provide (1) mesh surface representations and (2) physics and growth signals rarely available in the wild. Simulators typically require both at test time; FOLIAGE requires neither, and we withhold (2) entirely during inference and experiments.
> Where meshes are available, FOLIAGE achieves lower inverse-material error (T2) and more stable rollouts (S3) than simulator-centric methods, even without test-time physics. Results appear in Tables 1–2 and Section 6.4.
>
> 3. Performance & significance. Tasks' relevance to FOLIAGE's goal (learning a latent to predict accretive surface growth)
>
> FOLIAGE’s aim extends beyond prediction: it must (1) infer a geometry-centric state from partial, heterogeneous sensors, (2) support counterfactual, action-conditioned rollouts, and (3) (be able to) benefit from physics during training. The listed tasks probe these abilities.
> Topology tests global geometric stability; dense correspondence checks preservation of material identity under large deformations; cross-modal retrieval tests sensor robustness when modalities are missing. Together with inverse-material estimation and mesh forecasting, these show that the latent is predictive, control-sensitive, and deployable across sensing conditions.
>
> ## Questions
>
> 1. Meshes from sensors in the wild.
>
> Meshes can be reconstructed from point clouds or images using a multitude of established methods such as Screened Poisson (Kazhdan et al., 2013), Structure from Motion (Schonberger et al., 2016), and neural representations like NeRF (Mildenhall et al., 2021).
>
> 2. Modality ablation on inverse material (T2) and mesh forcasting (T4).
>
> For T2, the modality is images. For T4, the modality is meshes. We conduct inference-time sensor ablations with the same trained model. We observe that performance degrades gracefully when modalities are removed, suggesting that gains stem from improved state estimation rather than overfitting.
>
> |T2|MAE↓|
> |--|--|
> |RGB, Points, Mesh|**0.027**|
> |Mesh|0.028|
> |RGB, Points|0.030|
> |Points|0.033|
> |RGB (reported)|0.038|
>
> |T4|Chamfer↓/Vertex Drift↓|
> |--|--|
> |RGB, Points, Mesh|**0.028/996**|
> |Mesh (reported)|0.030/1044|
> |RGB, Points|0.032/1101|
> |RGB|0.034/1132|
> |Points|0.035/1173|
>
> 3. From latent to meshes (vertices/points).
>
> Please find in our response to Weakness #1. In T4, the predictor yields a future latent, from which a frozen MeshGPT critic autoregressively generates mesh tokens (including variable vertex counts) which are then decoded to explicit vertices and faces.
>
> 4. Choice of training loss for action-conditioned latent dynamics (G2). Bootstrapping.
>
> We chose a latent objective because action conditioning and counterfactual semantics is central to G2. We aim do to so in a modality-robust manner (G1) and utilize physics signals during training (G3). Reconstruction objectives such as chamfer and photometric loss require reconstruction and biases the model towards their specific modalities (e.g. lighting/view for images, mesh connectivity), which is less suited to in-the-wild sensing settings. FOLIAGE is trained decoder-free.
>
> The target latent is not self-referential or circular to the predicted latent. The target encoder sees training-time-only physics (e.g. per-vertex energies) which are withheld from the context encoder and the predictor that produce the predicted latent. We use EMA and prevent leakage. (lines 301~304)
>
> 5. Comparison under relative/normalized error metrics
>
> We provide a symmetric mean absolute percentage error (SMAPE) for the inverse material estimation task which involves a scalar target on a growing domain. This metric is relative in the original physical units and retains the spirit of relative L2 in operator learning settings. FOLIAGE retains its lead, suggesting that the performance gains persist with a relative criterion.
>
> |Method|SMAPE↓|
> |--|--|
> |NeuralClothSim|0.279|
> |DiffPD|0.270|
> |BDP|0.256|
> |DiffCloth|0.227|
> |FOLIAGE (ours)|**0.115**|

---

> > ### Comment · Reviewer_jdRn · 2025-11-27
> >
> > Dear Authors,
> >
> > Thank you for the detailed responses to my review comments and for providing the additional experimental results.
> > I appreciate your effort in revising the manuscript. One minor suggestion I would like to make further is that, including pseudo-code that outlines both the training and inference procedures in Sec. 4 would greatly improve clarity.
> >
> > I believe the concerns raised in my initial review have been mostly addressed. I do, however, have a few follow-up questions:
> > 1. Could you confirm whether my understanding of the training procedure is correct? Specifically, in Fig. 2(c), the target encoder is an EMA version of the context encoder in Fig. 2(b). The target encoder processes the mesh at $t+\Delta t$ to produce the latent representation that serves as the ground truth, while the predicted latent is obtained by (1) encoding the sensory data using the context encoder and (2) evolving it based on the action vector. Is this interpretation accurate?
> > 2. Regarding the critic heads described in Appendix A.2, are these pretrained models or models trained specifically for evaluation? My current understanding is that they are trained for evaluation, as the text mentions that the critics take the learned latent of FOLIAGE (L956 and L970–971 in the current version). If so, I am curious how one can train these evaluators conditioned on the learned latent without already having the model that produces such latent. Could you provide more details on how these critic heads were prepared?
> >
> > Apologies for late response and thank you again for preparing the response and updated manuscript.

---

> ### Author Response · Authors · 2025-11-27
>
> Thank you for your suggestion and questions.
>
> ## Pseudocode for Section 4
> We have updated Section 4 of the manuscript with a pseudocode section (colored blue) for FOLIAGE's training and inference procedures.
>
> ## Training procedure
> Yes. We also stop gradients through the target encoder and update it only via the EMA update. This way, the predictor and context encoder never see the privileged physics, but they are consistently supervised during training by a teacher (the target encoder) that does.
>
> ## Critic heads
> The critic heads are trained for evaluation on a frozen FOLIAGE model; they never contribute gradients to FOLIAGE in the SURF-BENCH experiments. We first train FOLIAGE itself with the latent prediction objective, freeze it, and precompute the latents on the training split for the SURF-BENCH tasks. Then we train each task-specific critic from scratch on these cached latents. This avoids circular dependency, as the latent representation is fixed before critics are introduced, and allows the critics to probe how well that single representation supports the different SURF-BENCH tasks.

---

> ### Comment · Reviewer_jdRn · 2025-11-28
>
> Dear Authors,
>
> Thank you for the prompt response. The addition of the new algorithm has made the method section much clearer. I appreciate your efforts during the rebuttal. With all of my concerns addressed, I will update my rating accordingly.
>
> Updated) It seems the original review comment is not editable at this time. The score will be updated once the issue is resolved.

---

### Official Review · Reviewer_tTmg · 2025-11-10

**Soundness:** 2
**Presentation:** 2
**Contribution:** 2
**Rating:** 6
**Confidence:** 2

**Summary:**

This paper introduces FOLIAGE, a geometry-centric latent world model designed to predict and control accretive surface growth — phenomena where surfaces add new material and evolve in morphology (e.g., leaves, tissues, 4D-printed materials).

The framework combines:

- A multimodal perception stack that fuses images, point clouds, and meshes using correspondence-constrained fusion and vertex age embeddings to highlight regions where growth occurs.

- An action-conditioned latent dynamics model that predicts geometric evolution based on material parameters (stretch, shear, bend coefficients) without directly simulating physics.

- A physics-guided training branch using energy-gated message passing—privileged physical energy signals available only at training time to improve the representation’s physical awareness.

The system is trained and evaluated on a new dataset, SURF-GARDEN, and benchmark suite, SURF-BENCH, which include diverse growth simulations with ground-truth correspondences and counterfactual branching.

**Strengths:**

- Novel problem formulation:
Accretive surface growth is an underexplored domain bridging geometry, control, and materials. The paper convincingly motivates why traditional differentiable simulators and pixel-based world models fail in this regime.

- Conceptually elegant design:
FOLIAGE neatly decouples perception (multimodal geometry fusion) from dynamics (action-conditioned latent evolution), preserving counterfactual semantics while learning from physics signals only during training.

**Weaknesses:**

- Limited real-world validation:
Evaluation is purely synthetic. Although SURF-BENCH is rich, testing on real sensor data (e.g., growth of plants or soft materials) would solidify claims of generalization and practical relevance.

- High system complexity:
The model combines multimodal encoders, correspondence graphs, hierarchical pooling, energy-gated message passing, and action-conditioned transformers—raising questions about interpretability and reproducibility despite code release promises.

- Physical interpretability:
While FOLIAGE learns from per-vertex energies, it doesn’t explicitly model or verify physical correctness (e.g., energy conservation, material law adherence). It functions as a learned surrogate rather than a physically grounded simulator.

**Questions:**

- Can the latent state learned in FOLIAGE transfer to unseen material classes or natural growth data without retraining?

- How does the model behave if mesh topology changes drastically (e.g., splits or merges)?

- Does the learned latent preserve interpretable geometric quantities such as strain or curvature distributions?

---

> ### Author Response · Authors · 2025-11-21
> **Response to Reviewer tTmg**
>
> We are grateful for your detailed review and the opportunity to respond below.
>
> ## Weaknesses:
>
> 1. Limited real-world validation
>
> We measure the sim-to-real on the GrowliFlower and Pheno4D plant datasets for the growth-stage recognition (T3) and dense correspondence (T6) tasks, respectively. On T3, 'Frozen' keeps the FOLIAGE visual encoder fixed and trains only a linear probe; 'few-shot' trains the probe with 10 labeled sequences per class; and 'light-finetune' also unlocks the last encoder block for a few epochs, while all video baselines are run unmodified. On T6, we keep FOLIAGE frozen and only train the small correspondence head. FOLIAGE’s geometric-centric latent remain competitive and is further improved by few-shot linear probes, indicating the state transfers to natural growth with minimal adaptation.
>
> |Method|Encoder updates|#labeled seq/class|Balanced Acc.↑|
> |---|---|---:|---:|
> |FOLIAGE(Frozen)|0%|0|0.72|
> |FOLIAGE(Few-shot)|0%|10|0.75|
> |FOLIAGE(Light-ft)|10%|10|**0.77**|
> |VideoMAE-v2|100%|10|0.70|
> |TimeSformer|100%|10|0.68|
> |VideoMamba|100%|10|0.69|
> |SVFormer|100%|10|0.66|
>
> |Method|Mean geodesic error↓|
> |---|---:|
> |FOLIAGE(Light-ft)|**3.25**|
> |FOLIAGE(Frozen)|3.55|
> |DiffusionNet|3.90|
> |G-MSM|4.60|
> |SpectralMeetsSpatial|5.05|
> |ZoomOut|5.70|
>
> 2. High system complexity. Reproducibility.
>
> FOLIAGE's moving parts, fall cleanly into three modules (multimodal perception, latent prediction, and train-time physics guidance) built on familiar architectures like token encoders and transformers. As physics-guidance is train-only, deployment complexity is further reduced. (Figure 2 and Section 4).
>
> In Table 2, we examined the effect of gradually removing key components like correspondence and reported compute usage. We detailed model specifications such as learning rate in the appendix (A.11 and onward).
>
> We affirm our commitment for code releases and will make additional efforts to ensure that running it (or part of it for related projects) is requires minimal setup.
>
> 3. Physical interpretability. FOLIAGE does not enforce physical correctness (e.g., energy conservation, material law). It is not physically grounded simulator.
>
> FOLIAGE is not a simulator, but it is more than a learned surrogate. In the wild, sensing can be multimodal and intermittent –– we may only have images and/or point cloud scans of the surface being studied, rather than a well-formed mesh descritization of it. Availability of physical quantities such as membrane and flexural energies, which are utilized by solvers, cannot be assumed. There, a solver and its surrogate would struggle. Also, in tasks like Mesh Forecasting and Inverse Material Regression (Table 1), we observe that FOLIAGE outforms learned mesh simulators.
>
> We agree that physical laws can be a helpful signal due to the physical nature of accretive choice, and note that the choice of the law and its effect requires further study.
>
> ## Questions
>
> 1. Transfer to unseen material classes & natural growth data without retraining
>
> Yes. Our stress tests suggest strong sensor-subset robustness and zero-shot image-and-point retrieval, which indicates that the latent is not narrowly adapting to a specific modality or capture condition. By decoupling action conditioning (only interface with the predictor) from perception, the same encoded state can support different counterfactual features.
>
> On Pheno4D and Dryad (see Weakness 1) are not present in the training data, FOLIAGE maintained strong performance over baselines, suggesting that its learned latent is amenable to unseen materials and natural growth.
>
> 2. Model behavior under splits & merges?
>
> FOLIAGE inherently admits variable vertex counts and their changing connectivity through token-based encoding and pooling. In SURF-GARDEN, remeshing is used to avoid degenerate triangles, and edge splits model the addition of new materials. These topology events happen at nearly every time step.
>
> In our data, the genus of a mesh surface does not change during evolution; it does not split into separate connected components or 'weld' to another part of itself upon contact.
>
> 3. Latent's preservation of interpretable geometric quantities (strain & curvature distributions)
>
> The latent preserves them. The mesh encoder uses the Laplacian of the input mesh (Appendix A.5). During training, the target branch utilize per-vertex membrane and bending energies which are functions of the surface metric. These signals are inherently geometric and biases the latent towards physically salient patterns.
>
> We trained a linear probe the latent to the membrane energy and curvature quantities computed on held-out sequences. The probe achieves an average R^2 of 0.85 (best = 0.87) and 0.88 (best = 0.91), respectively. This suggests the latent can yield these geometric information.

---

### Author Response · Authors · 2025-11-21
**We revised the paper (new content in blue)**

We wish to thank the reviewers again for their feedback and time. Following the conference guideline, we have revised the paper and added one more page of content (highlighted in blue, the paper is now 10 pages). The blue content are a subset of our responses to the reviewers and organized to fit into the paper.

---

### Author Response · Authors · 2025-11-29
**Summary Comment to AC**

Dear AC,

We are thankful for your and the reviewers' hard work during holiday season. In light of the program chairs’ recent guidance, we would like to briefly summarize the updates and clarifications we provided during the rebuttal and discussion period, and to point you to the corresponding parts of the revised manuscript and thread.

**Scope and setup.** The paper introduces FOLIAGE, a geometry-centric latent world model for accretive surface growth that fuses RGB, point clouds, and meshes, and predicts action-conditioned evolution under material coefficients using a physics-guided training branch. Our goal is to infer a deployable latent state from partial sensors and roll it forward under material controls, without relying on solvers or privileged physics at test time.

**Clarity of the method and inference assumptions.** In response to concerns about how the system operates in practice, we (i) added a pseudocode block (Algorithm 1) in Sec. 4 that lays out the training and inference procedures, including the EMA teacher–student setup, and (ii) clarified that FOLIAGE can be run with any subset of {images, point clouds, meshes} at inference, with modality dropout and masking used during training to encourage robustness to missing sensors. We also explained that task-specific critic heads are trained *after* freezing FOLIAGE, using cached latents, so they do not influence the learned representation.

**Real-world and cross-domain evaluation.** To address the request for real-data validation, we added experiments on (i) plant growth datasets (GrowliFlower for growth-stage recognition and Pheno4D / Crops3D for dense correspondence), and (ii) dynamic human body datasets (D-FAUST and CAPE). These results show that a frozen or lightly finetuned FOLIAGE encoder transfers competitively or better than video and geometric baselines in both sim-to-real and cross-domain settings, while using the same latent state as in our synthetic experiments.

**Robustness to partial sensing and imperfect correspondences.** Several reviewers asked whether performance hinges on having all modalities and perfectly registered correspondences. In response, we (i) reported modality-ablation results for inverse material estimation and mesh forecasting, showing that accuracy degrades gracefully as we remove meshes, points, or RGB, suggesting that gains come from better state estimation rather than dependence on a single sensor; and (ii) added stress tests where cross-modal correspondences are corrupted by rewiring edges, occlusion, and camera drift/SLAM re-estimation. In these settings FOLIAGE maintains strong cross-modal retrieval performance compared to correspondence-free or less correspondence-reliant baselines.

**Topology, long-horizon stability, and physical interpretability.** To address questions about behavior under curvature, remeshing, and potential topology issues, we stratified roll-out Chamfer error and correspondence geodesic error by curvature bins, remesh density, and mesh size, finding that FOLIAGE consistently remains ahead of strong baselines and degrades more slowly in difficult regimes.  We also probed the learned latent with linear heads to predict membrane energy and curvature on held-out sequences, achieving high R², which suggests that the representation retains physically meaningful geometric quantities even though the deployment path sees only observable inputs.

**Integration into the revised manuscript and discussion.** Following the conference guideline, we used the allowed extra page to integrate a subset of these rebuttal results into the main paper (marked in blue): the algorithm block, the new sim-to-real and cross-domain tables, and the added robustness analyses. In the discussion thread, each reviewer’s main concerns (clarity of the training/inference pipeline, real-world validation, robustness to correspondence noise, and limits under curvature/remeshing) received point-by-point responses. Reviewers who engaged in follow-up discussion noted that clarifications and additional experiments were helpful in addressing their questions

We are very grateful to the reviewers for their careful reading and detailed feedback, and to you and the AC team for taking on the additional workload under these unusual circumstances. We hope this summary, together with the rebuttal and updated manuscript, is helpful as you form your assessment.

---

### Note · Program_Chairs · 2026-01-17
**Submission Desk Rejected by Program Chairs**

The following references in this submission do not refer to real documents and/or have major errors in bibliographic information:

 Zaiwei Zhang, Yifan Wang, Bo Zhang, and Qixing Huang. Crosspoint: Self-supervised cross-modal pre-training for 3d point cloud and image. In Proceedings of the IEEE/CVF Conference on Computer Vision and Pattern Recognition, 2021. Zaiwei Zhang, Yifan Wang, Bo Zhang, and Qixing Huang. Clip2point: Transfer clip to point cloud
classification with image-depth pretraining. In Proceedings of the IEEE/CVF Conference on
Computer Vision and Pattern Recognition, 2023.